



# Automated precipitation monitoring with the Thies disdrometer: Biases and ways for improvement

Michael Fehlmann[1], Mario Rohrer[1,2], Annakaisa von Lerber[3], and Markus Stoffel[1,4,5]

[1]Climate Change Impacts and Risks in the Anthropocene (C-CIA), Institute for Environmental Sciences, University of Geneva, Geneva, Switzerland
[2]Meteodat GmbH, Zurich, Switzerland
[3]Finnish Meteorological Institute (FMI), Helsinki, Finland
[4]Department of Earth Sciences, University of Geneva, Geneva, Switzerland
[5]Department F.-A. Forel for Environmental and Aquatic Sciences, University of Geneva, Geneva, Switzerland

**Correspondence:** Michael Fehlmann (michael.fehlmann@unige.ch)

**Abstract.** The intensity and phase of precipitation at the ground surface can have important implications for meteorological and hydrological situations, but also in terms of hazards and risks. In the field, Thies disdrometers are sometimes used to monitor the quantity and nature of precipitation with high temporal resolution and very low maintenance and thus provide valuable information for the management of meteorological and hydrological risks. Here, we evaluate the Thies disdrometer

with respect to precipitation detection as well as the estimation of precipitation intensity and phase at a pre-alpine site in Switzerland (1060 m a.s.l.), using a weighing precipitation gauge (OTT pluviometer) as well as a two-dimensional video disdrometer (2DVD) as a reference. We show that the Thies disdrometer is well suited to detect even light precipitation, reaching a hit rate of around 95%. However, the instrument tends to systematically underestimate rainfall intensities by 16.5%, which can be related to a systematic underestimation of the number of raindrops with diameters between 0.5 and 3.5 mm. During

snowfall episodes, a similar underestimation is observed in the particle size distribution (PSD), which is, however, not reflected in intensity estimates, probably due to a compensation by snow density assumptions. To improve intensity estimates, we test PSD adjustments (to the 2DVD) as well as direct adjustments of the resulting intensity estimates (to the OTT pluviometer), which are both able to reduce the systematic deviations during rainfall. For snowfall, the combination of the 2DVD and the OTT pluviometer seems promising as it allows improvement of snow density estimates, which poses a challenge to all optical

precipitation measurements. Finally, we show that the Thies disdrometer and the 2DVD agree well insofar as the distinction between rain and snowfall is concerned, such that an important prerequisite for the proposed correction methods is fulfilled. Uncertainties mainly persist during mixed phased precipitation or low precipitation intensities, where the assignment of precipitation phase is technically challenging, but less relevant for practical applications. We conclude that the Thies disdrometer is not only suitable to estimate precipitation intensity, but also to distinguish between rain and snowfall. The Thies disdrome-

ter therefore seems promising for the improvement of precipitation monitoring and the nowcasting of discharge in pre-alpine areas, where considerable uncertainties with respect to these quantities are still posing a challenge to decision making.



# 1 Introduction

The intensity and type of precipitation falling on the ground surface (e.g. rain, snow, drizzle, or hail) often determines the absence or occurrence of subsequent processes. A detailed knowledge on the nature and intensity of precipitation is therefore

decisive in terms of hazards and ensuing risks. For example, for the management of traffic roads, it is important to know whether falling snow or mist will likely hamper road conditions or visibility (Toivonen and Kantonen, 2001). For example, Juga et al. (2012) show that very poor visibility due to intense snowfall combined with reduced road surface friction caused a severe flow of accidents in Helsinki on 17 March 2005. Likewise, the occurrence of freezing rain at the ground surface can lead to the collapse of trees and power supply lines with potentially catastrophic cascading effects, as was experienced during

a recent case in Slovenia (Kämäräinen et al., 2017; Schauwecker et al., 2019). Last not least, both precipitation intensity and its phase (i.e. rain or snowfall) are decisive for runoff formation and the occurrence of flash floods in (pre-)alpine catchments (e.g., Fehlmann et al., 2018; Tobin et al., 2012).

To support decision making and intervention in such situations, the Thies Clima laser precipitation monitor (in the following referred to as Thies disdrometer) offers the possibility to measure precipitation intensity and type with a high temporal

resolution; the monitor can therefore replace present weather observations from manned stations to a certain degree (Merenti-Välimäki et al., 2001). Due to their low maintenance requirements, disdrometers have ben used widely for operational weather monitoring for road or air traffic. In the future, disdrometers will likely be employed more often for hydrological purposes as well, with the aim of monitoring heavy precipitation and the ensuing nowcasting of river discharge, particularly in mountainous environments where precipitation phase estimates are still uncertain (e.g., Unterstrasser and Zängl, 2006). Although the

Thies disdrometer has been tested previously with the aim of verifying dual-polarimetric weather radars, and in particular their hydrometeor classification algorithms (Pickering et al., 2019), not many studies have assessed uncertainties of disdrometer measurements so far. Furthermore, weather radars still suffer from limitations in the detection of convective precipitation or due to the blocking of the radar signal at lower elevations by mountain topography (Besic et al., 2016), therefore rendering reliable ground observations even more important in these areas.

In this study, we evaluate the Thies disdrometer with respect to precipitation detection as well as the monitoring of precipitation intensity and phase at a well-instrumented measuring site in Switzerland (Innereriz, 1060 m a.s.l.). To this end, we have used a weighing precipitation gauge (OTT pluviometer) as well as a two-dimensional video disdrometer (2DVD) as reference instruments over a measurement period of two years. The 2DVD provides accurate information about the volume and velocity of falling hydrometeors and has already been used previously as a reference to correct particle size and velocity (Parsivel) dis-

tributions of laser disdrometers during either rainfall (Leinonen et al., 2012; Raupach and Berne, 2015) or snowfall (Battaglia et al., 2010). In this study, we include both solid and liquid precipitation events and point to differences in resulting correction methods. Furthermore, we develop a hydrometeor classification algorithm for the 2DVD measurements as a basis for the evaluation of precipitation phase estimates. Whereas other studies have developed such algorithms using bulk variables for the





classification (e.g., Grazioli et al., 2014), we have implemented here a particle-by-particle classification method allowing to explore resulting mixing ratios in the case of mixed-phased precipitation.

The paper is organized as follows: In section 2, the measurement devices are presented in more detail and the processing of the raw data is described. In section 3, biases and proposed corrections of the Thies disdrometer are presented with respect

5 to precipitation detection as well as the monitoring of precipitation intensity and phase. Results are discussed in section 4, before conclusions with respect to the operational monitoring of precipitation with the Thies disdrometer as well as potential applications in a hydrological context are drawn in section 5.

## 2 Data and methods

### 2.1 Measurement devices

10 The Thies disdrometer is evaluated in this study by using a weighing precipitation gauge (OTT pluviometer) and a 2DVD as a reference. Measurements have been taken over a duration of two years (2017-07-01–2019-06-30). These instruments have been set up at Innereriz, Switzerland (1060 m a.s.l., Fig. 1) and are described in more detail in the following.

The Thies disdrometer is designed to estimate precipitation intensity as well as different types of precipitation (e.g. drizzle, rain, hail, snow or mixed precipitation). Precipitation type and intensity are estimated on the basis of an optical principle,

15 i.e. by the generation of a laser beam (786 nm) attenuated by falling particles (Fig. 2, left). The strength and duration of this attenuation allows inference of the diameter and velocity of the falling particles, such that precipitation type can be estimated by using empirical relationships between these two quantities (e.g., Gunn and Kinzer, 1949). To derive precipitation intensity from raw particle data, several assumptions have to be made, also regarding particle shape and density. Whereas for liquid precipitation, an oblate shape (Chuang and Beard, 1990) and a density of 1 $g/cm^3$ is assumed, a spherical shape is considered

20 for solid precipitation. The density of a snow particle (ranging from 5–450 $g/cm^3$) is estimated taking its diameter and velocity as well as the ambient temperature into account. As the exact relationship used is not reported by the manufacturer, a simplified relationship between particle diameter and density is derived in this study to estimate precipitation intensities in case of snowfall (Section 3.2). The dominant precipitation type (WMO table 4680) as well as precipitation intensity is reported by the instrument every minute. Furthermore, particle diameter and velocity distributions are summarized by the number of particles recorded in

25 paired classes of diameters (20 classes, ranging from 0.125–9 mm) and velocity (22 classes, ranging from 0–12 m/s), yielding a total of 440 classes.

The 2DVD, developed by Joanneum research, is based on a principle similar to that of the Thies disdrometer, but the 2DVD is able to derive more direct and more detailed information about individual hydrometeors. Maintenance requirements for the instrument are not negligible and it is mainly used in the research context and in combination with radar observations (e.g.,

30 Bringi et al., 2015; Gorgucci and Baldini, 2015; Huang et al., 2010, 2015; Thurai et al., 2012). Furthermore, the 2DVD has been used for the correction of laser disdrometers (Raupach and Berne, 2015). As shown in Fig. 2 (right), falling hydrometeors are detected by two optical cameras from two perspectives, which allows to derive more detailed information about the shape and volume of individual particles as well as about their velocity. Information about these quantities is reported for each individual





particle, including the exact time of the observation (in ms). Precipitation type is not (yet) reported by the instrument, but can be estimated on the basis of the raw particle data. To validate precipitation type estimates by the Thies disdrometer, a classification algorithm was developed in this study, allowing an estimation of the type of each individual hydrometeor (Section 2.2).

The OTT pluviometer is designed to automatically determine precipitation intensities and amounts. Unlike tipping bucket
rain gauges, this instrument is based on the weighing of the precipitation amount in a high-precision load cell. Advantages compared to a tipping bucket rain gauge are particularly related to the measurement of solid precipitation amounts, resulting in fewer losses due to the evaporation as well as the avoidance of a temporal lag effect in the measurement (Savina et al., 2012). The instrument is thus able to measure precipitation amounts with high accuracy and is therefore used as a reference for precipitation amounts at the ground surface in various applications, including the validation of disdrometers (e.g., Raupach
and Berne, 2015). A well-known problem when using precipitation gauges mounted above ground is, however, the undercatch due to the influence of wind, which has been extensively studied for rainfall (e.g., Pollock et al., 2018) and in particular for snowfall (e.g., Fassnacht, 2004; Kochendorfer et al., 2017; Yang, 2014; Wolff et al., 2015). The undercatch is thereby found to be larger for snowfall than for rainfall and to increase with increasing wind-speed. In this study, however, we do not explicitly correct for wind effects as wind speeds at the study site are generally very low (on average 0.46 m/s during the investigated
time period). The maintenance requirements of the instrument are relatively low - only the container, which holds 750 mm for the model used, must be emptied regularly. Precipitation intensity and amount are reported every minute.

Finally, temperature and wind measurements of a LUFFT weather sensor are used in this study. This sensor was located at the same measuring station (Fig. 1, left) and provided corresponding measurements every minute. Temperature is measured by way of a highly accurate NTC-resistor in a ventilated housing with radiation protection in order to keep the effects of external
influences (e.g. solar radiation) as low as possible. The wind meter uses 4 ultrasonic sensors which take cyclical measurements in all directions. The resulting wind speed and direction are calculated from the measured run-time sound differential.

## 2.2   2DVD classification algorithm

As precipitation type is not reported by the 2DVD by default, a classification algorithm was developed in this study, to assign one of the following precipitation types to each observed hydrometeor: hail, rain, melting snow, graupel or snow. Unlike other
algorithms (e.g., Grazioli et al., 2014), the algorithm used here is based on a particle-by-particle classification rather than on bulk information, which even allows for the explicit quantification of hydrometeor mixtures during a given time period. For the validation of the Thies disdrometer, the dominant precipitation type during 1-min observations was then estimated on the basis of these mixing ratios.

Similar to the Thies disdrometer, the classification algorithm is based on the empirical relationship between particle diameter
D and fall velocity V, which varies among different types of precipitation. The equations used (equations 1–5) are based on literature (Gunn and Kinzer, 1949; Locatelli and Hobbs, 1974; Mitchell, 1996) as well as measurements and analyses conducted





by Joanneum Research.

$$V_{Hail} = 3.74 \cdot D^{0.5} \tag{1}$$

$$V_{Rain} = 9.65 - (10.3 \cdot e^{-0.6 \cdot D}) \tag{2}$$

$$V_{Melting} = 4.65 - (5 \cdot e^{-0.95 \cdot D}) \tag{3}$$

$$V_{Graupel} = 1.3 \cdot D^{0.66} \tag{4}$$

$$V_{Snow} = 0.79 \cdot D^{0.24} \tag{5}$$

A particle is considered for classification only if its diameter D and velocity V lie within a valid range (0.6 mm < D < 9 mm, V < 17 m/s) and if no major differences exist in particle size between the two cameras (0.8 < $H_A$/$H_B$ < 1.25, where $H_A$ and $H_B$ denote the particle height in camera A and B, respectively). For each valid particle, theoretical fall speeds for different precipitation types are calculated according to its diameter and equations 1–5. The estimated values are then compared to measured velocity, whereas precipitation type is determined according to the closest match between these values. In addition, snow or melting snow above 10 °C is reclassified as rain - a plausibility check which is also applied by Thies Clima for the processing of Thies disdrometer data. An example of the resulting particle-by-particle classification is given in Fig. 3 for a transition from rain to snowfall during 6 h.

After the inspection of 1-min mixing ratios of different precipitation types obtained by this classification algorithm (not shown here), we determined the dominant precipitation phase during 1 min as follows (Table 1): Rain is considered dominant if more than 70% of the particles are classified as rain, whereas snow and/or graupel are considered dominant if more than 80% of the particles are classified as snow, melting snow or graupel. Furthermore, hail is already assigned for mixing ratios greater than 1% as the chance of (larger) hailstones being captured by the relatively small measuring area is quite small. In the remaining cases, mixed-phased precipitation is assigned.

### 2.3 Comparison of measurements and performance measures

The Thies disdrometer is evaluated by using the OTT pluviometer as a reference for precipitation detection and intensities and the 2DVD as a reference for PSD and precipitation type. The following comparisons refer to a time period of two years (2017-07-01–2019-06-30) during which all these instruments have been installed simultaneously. Whereas the first year of measurements (2017-07-01–2018-06-30) is used for the design of the proposed correction methods, the second year of measurements (2018-07-01–2019-06-30) is used for the independent validation of the methods.

When comparing the Thies disdrometer with the OTT pluviometer, corresponding 1-min observations are merged. Although both instruments are measuring with a resolution of 1 min, they have not been set up to measure synchronously. To avoid mismatches due to temporal shifts between observations, the minimum integration time considered for the evaluation of precipitation detection and precipitation intensities was set to 5 min. As the effect of increasing integration time on the reliability of measurements can be of interest for operational applications, we also report results for integration times up to 4 h (i.e. 5, 10, 20, 30, 60, 90, 120 and 240 min). Intensities for different integration times are calculated based on the cumulative precipitation





sum, which is given by both instruments. For all correction methods applied, the variable of interest is first integrated over the considered integration time before any correction is applied.

When comparing the Thies disdrometer (or the OTT pluviometer) with the 2DVD, 1-min observations can be used and the 2DVD data is aggregated accordingly. When comparing the PSD between the two disdrometers, the number of particles is
normalised by the so-called effective measuring area. This area slightly deviates from the actual measuring area (being 45.32 $cm^2$ for the Thies disdrometer and 109.39 $cm^2$ for the 2DVD) as a function of particle diameter. Essentially, the effective measuring area decreases for larger particles due to the increasing non-recognition of partially observed particles at the border of the measuring area. Whereas the effective measuring area is reported by the 2DVD for each observed particle, it is calculated for each diameter class of the Thies disdrometer after Angulo-Martínez et al. (2018), using the mean diameter of each class.
For the adjustment of the particle size distribution (PSD) measured by the Thies disdrometer, we adopt a method proposed by Raupach and Berne (2015), which essentially scales drop concentrations per diameter class to ensure that they on average match those recorded by the 2DVD. For the consistent comparison of precipitation phase between the two disdrometers, certain precipitation types were aggregated according to Table 1.

For the evaluation of categorical variables, i.e. precipitation detection (yes/ no) and precipitation phase (rain/ mixed/ snow),
hit and false alarm rates with respect to the reference instrument are calculated according to Jolliffe and Stephenson (2012). For the evaluation of precipitation intensity, systematic deviations are given in terms of the absolute bias B (equation 6) and the scatter is described in terms of the correlation coefficient Corr (equation 7), where $\hat{x}$ denotes the estimation of the Thies disdrometer and x denotes the measurement of the OTT pluviometer for all observations n.

$$B = \frac{1}{n} \sum_{i=1}^{n} \hat{x}_i - x_i \tag{6}$$

$$Corr = \frac{cov(\hat{x}, x)}{\sqrt{var(\hat{x})var(x)}} \tag{7}$$

## 3  Results

### 3.1  Precipitation detection

The capacities of the Thies disdrometer to detect precipitation are assessed with the OTT pluviometer as a reference. After exploring the full time series, data from the first year of measurements was used to optimize precipitation detection by estab-
lishing minimum precipitation thresholds. The application of these thresholds was then evaluated during the second year of independent measurements.

Hit and false alarm rates of the Thies disdrometer with respect to precipitation detection are indicated with circles in Fig. 4 (left) for the whole time series and for different integration times. Thereby, hit rates are stable and reach values between 95.2 and 95.9%. False alarm rates are low for short integration times (e.g. 5.1% for periods of 5 min) but tend to increase with
increasing integration time (e.g. 14.1% for periods of 4 h). This increase is probably related to the fact that - given a dry period - the chance of misinterpreting a signal as precipitation is increasing with increasing integration time.





To reduce false alarm rates of the Thies disdrometer for longer integration times, we also tested the application of minimum precipitation thresholds. The receiver operating characteristic (ROC) curves shown in Fig. 4 (left) thereby depict all possible combinations of hit and false alarm rates that can be achieved by the introduction of such a threshold. The application of a minimum threshold will generally reduce both false alarm as well as hit rates. Therefore, an optimal threshold was defined for

each integration time by minimising the Euclidean distance to the upper left corner in the ROC diagram (i.e. to the theoretical optimum with a hit rate equal to 1 and a false alarm rate equal to 0), resulting in a balanced solution between the two measures. This optimization was applied to the first year of measurements and the resulting thresholds are listed in Table 2 for each integration time. Noteworthy, these thresholds (expressed in mm/h) are quite stable for different integration times with a mean of 0.04 mm/h.

The effect of applying the proposed thresholds on hit and false alarm rates during the second year of measurements is depicted in Fig. 4 (right) and Table 2. The application of such thresholds is particularly beneficial for integration times exceeding 20 min, as they allow to effectively reduce false alarm rates by up to 8.9 % (for periods of 4 h). For integration times shorter than 20 min, the application of a minimum precipitation threshold only has a negligible effect. Furthermore, by applying the proposed thresholds, a balanced solution with respect to hit rates and false alarm rates can be found for all the integration times

considered, resulting in a relatively similar distance to the theoretical optimum in the ROC diagram.

Finally, by applying the proposed minimum precipitation thresholds in Table 2, we analyze missed events as well as false alarms produced by the Thies disdrometer in the validation period in more detail, i.e. with respect to precipitation intensity and phase. Whereas precipitation intensities measured by the OTT pluviometer were of interest during missed events, precipitation intensities indicated by the Thies disdrometer were analysed during false alarms. To investigate whether the phase of precip-

20 itation could be relevant for missed events or false alarms, observations were separated according to a temperature threshold of 1.2 °C (Fehlmann et al., 2018). The resulting distributions of precipitation intensities and phase during missed events and false alarms are shown in Fig. 5. Precipitation intensities during missed events are decreasing with increasing integration time, mean intensities being around 0.6 mm/h during periods of 5 min and decreasing to around 0.03 mm/h during periods of 4 h. While precipitation intensities during missed events are very similar above and below the temperature threshold of 1.2 °C,

the relative frequency of missed events seems to be slightly higher below this temperature threshold. Precipitation intensities indicated by the Thies disdrometer during false alarms are very low, ranging from 0.12 mm/h for 5-min periods to 0.02 mm/h for 4-h periods, with no remarkable differences above and below the temperature threshold of 1.2 °C.

## 3.2 Precipitation intensities

The capacities of the Thies disdrometer to measure precipitation intensities were assessed with the OTT pluviometer as a

30 reference for precipitation intensities as well as with the 2DVD as a reference for the PSD. After exploring error patterns in the entire time series, the first year of measurements was used to establish corresponding correction methods. The application of the established correction methods was then evaluated with independent data from the second year of measurements.

Figure 6 depicts the cumulative precipitation sums of the Thies disdrometer over the full investigation period as compared to the OTT pluviometer. Precipitation sums for both instruments are separated into rain and snow according to 1-min precipitation



type estimates of the Thies disdrometer. As rain and snowfall events represent 89.5% of the total precipitation sum, we restrict analysis to these two precipitation types in the following. Total precipitation after two years of measurements is underestimated by 12.4% by the Thies disdrometer. This systematic underestimation is almost entirely related to rainfall events, during which the total precipitation sum is underestimated by even 16.5%. The underestimation during snowfall events is much smaller

(4.0%) and seems to be less systematic, but rather related to individual events during the second year of measurements.

As a first approach to improve precipitation intensity estimates by the Thies disdrometer, we tested a direct adjustment to the measurements of the OTT pluviometer. A comparison of precipitation intensities between these instruments during the full investigation period is shown in Fig. 7 for an integration time of 30 min; it confirms the error pattern described above: Whereas a systematic underestimation of rainfall intensities is visible, almost no systematic error can be seen with respect to snowfall

intensities. Furthermore, the systematic underestimation of precipitation intensities seems to be well captured by a constant factor (i.e. independent of integration time or precipitation intensity). We thus propose to use the mean ratio as a measure for the adjustment and to distinguish between rain and snowfall. Using the first year of measurements, the mean ratio for rain and snowfall intensities is 0.83 and 1.04, implying correction factors of 1.20 and 0.96, respectively.

As a second approach to improve precipitation intensity estimates by the Thies disdrometer, we tested an adjustment of the

15 PSD to the measurements of the 2DVD. A comparison of the PSD between these two instruments is shown in Fig. 8 for all rain and snowfall events. Although the overall shape of the PSD is similar for both instruments, systematic deviations seem to exist during both rain and snowfall events. During rainfall events, the number of particles with diameters between 0.5 and 3.5 mm (classes no. 4–12) is systematically underestimated, whereas the number of smaller and larger particles is overestimated by the Thies disdrometer as compared to the 2DVD. When looking at the monthly variability of the resulting correction

factors (Fig. 8, right), the overestimation seems to be less stable than the underestimation. During snowfall, the number of particles with diameters exceeding 0.75 mm is overestimated, whereas the number of smaller particles is underestimated as well. Noteworthy, the underestimation (classes no. 4–12 for rainfall and 5–22 for snowfall) will affect resulting estimates of precipitation intensity in particular. In the case of rainfall and assuming the mean PSD obtained by the Thies disdrometer, particles between 0.5 and 3.5 mm (classes no. 4–12) are contributing to 90% of the total rainfall volume. The smallest and

largest particles are almost negligible for total volume due to their small volume (smallest particles) and number (largest particles), respectively. Nevertheless, we propose to apply correction factors for the number of particles in each diameter class, and to further distinguish between rainfall and snowfall. Using the first year of measurements, the resulting correction factors for rain and snowfall are listed in Table 3. Given the corrected PSD, rainfall intensity is calculated by assuming a density of 1 g/cm$^3$. For snowfall, a relationship between particle diameter and density is established by comparing 1-min accumulated

volumes (measured by the 2DVD) to the corresponding mass (measured by the OTT pluviometer) and is shown in Figure 9.

The effect of both correction methods proposed here was subsequently tested during the second year of measurements. The resulting performance measures before and after the correction are given in Table 4 for different integration times. Performance measures are thereby calculated for the whole dataset as well as for all rain and snowfall separately. An example for the integration time of 30 min is further given in Fig. 10. As can be seen in Table 4 and Fig. 10, the performance of both correction

methods is comparable, whereas the main effect is the reduction of the bias for rainfall intensities. The correlation for both





rain and snowfall events can only be improved through an adjustment to the 2DVD, as the adjustment to the OTT pluviometer is based on a constant factor. Thereby, correlation is slightly improved regarding snowfall events but remains unchanged for rainfall events. This suggests that the adjustment of the PSD has a comparable effect than a linear adjustment of resulting precipitation intensities. During snowfall, however, the relation of particle densities to drop diameter can result in non-linear

effects and improve the correlation with respect to the OTT pluviometer. Uncertainties in precipitation intensities are higher during snowfall than during rainfall, and correlations generally increase with increasing integration time, with this increase being most pronounced when increasing the integration time from 5 to 20 min. This finding also indicates that at least some uncertainties in estimates of precipitation intensities (including small time shifts between observations) are averaged out over longer integration times.

## 3.3 Precipitation phase

The capacities of the Thies disdrometer to detect the predominant precipitation type is assessed using the 2DVD as a reference. We thereby focus on the precipitation phase, i.e. the distinction of rain and snowfall, which has been shown above to be an important criterion for the proposed correction methods. Furthermore, we only consider pairwise complete (1 min) observations of both instruments with either rain, snow or mixed precipitation, resulting in a time series of 2,533 h of precipitation.

Table 5 shows the agreement of precipitation phase estimates between the Thies disdrometer and the 2DVD during the full time series, while Fig. 11 depicts the relative frequency of observations as a function of temperature for both instruments, including an indication of the mixing ratios obtained by the 2DVD. Thereby, the Thies disdrometer agrees well with the 2DVD insofar as the classification of rain and snow is concerned. By contrast, larger differences exist with respect to the classification of mixed precipitation. Regarding the detection of rain, the Thies disdrometer reaches an almost perfect hit rate (99.7%).

However, the overall frequency of rain is slightly overestimated by the Thies disdrometer, being reflected by a false alarm rate of 9.9%. Regarding the detection of snow, the overall frequency of detected cases is almost equal for both instruments. The hit and false alarm rate of the Thies disdrometer with respect to the 2DVD is reaching 95.3 and 1.3%, respectively, reflecting a good agreement between the two instruments. Finally, the Thies disdrometer classifies much less cases as mixed precipitation (1%) than the 2DVD (4.3%), resulting in both a low hit and false alarm rate for these cases.

Most misclassifications are related to cases during which the Thies disdrometer indicates rain, whereas the 2DVD indicates mixed precipitation or snow. As can be seen in Fig. 11, such cases occur at both temperatures above and well below 0 °C. Thereby, the Thies disdrometer seems to overestimate cases of rain below 0 °C and to underestimate cases of snowfall or mixed precipitation above 0 °C as compared to the 2DVD. At least during distinct misclassifications, i.e. in cases where the Thies disdrometer indicated rain while the 2DVD indicating snow, it can be shown that precipitation intensities are very small,

i.e. their mean being 0.19 mm/h (while being 0.93 mm/h for all cases).



## 4   Discussion

In this study, we have shown that the Thies disdrometer is well suited for precipitation detection, reaching hit rates of around 95% with respect to the OTT pluviometer. False alarm rates might be affected by the sensitivity of the reference instrument, but are comparable to findings of Bloemink and Lanzinger (2005) who use human observations as a reference.

We have further demonstrated that the Thies disdrometer systematically underestimated rainfall intensities at the study site by 16.5% during two years of measurements, which we explain to be related to an underestimation of drop concentrations for drop diameters ranging between 0.5 and 3.5 mm. At the same time, larger and smaller drops are overestimated by the instrument; this is, however, less relevant for the resulting estimates of rainfall intensities. Other studies have reported similar patterns in terms of bias in the PSD and while analyzing other disdrometers, such as the Joss–Waldvogel (Leinonen et al.,

2012) or the OTT Parsivel (Raupach and Berne, 2015) disdrometers. However, the deviations in the PSD and implications for rainfall intensity estimates can be different between different types of instruments: For example, the OTT Parsivel disdrometer only underestimates drops with sizes ranging between 0.8 and 1.6 mm and only during periods of higher rainfall intensity. In addition, the device tends to even overestimate rainfall intensities (Raupach and Berne, 2015). Interestingly, an overestimation of rainfall intensities is also reported for the Thies disdrometer at the intercomparison site Wasserkuppe in Germany

(Lanzinger et al., 2006). Supposedly, this contrary result to our study is due to differences in wind exposure. While our study site in Innereriz is extremely wind sheltered (average wind speeds being 0.46 m/s during the investigation period), the site at Wasserkuppe is strongly exposed to wind, average wind speeds being 6.4 m/s from 1999–2018 (data obtained by German weather service DWD). Despite these differences, the correction of the PSD as proposed by Raupach and Berne (2015) could successfully be adopted in this study to reduce the bias found in rainfall intensity estimates as given by the Thies disdrometer.

The correlation coefficient remains mostly unaffected by the correction, which is again consistent with the results reported by Raupach and Berne (2015). Consequently, an analogous effect can be achieved by applying a linear adjustment to a weighing precipitation gauge, which is proposed as an alternative correction method in this study. It should be noted further that the overestimation of smaller drops by laser disdrometers with respect to the 2DVD is also found in other studies (Krajewski et al., 2006; Raupach and Berne, 2015), but can at least partly be related to unreliable estimates by the 2DVD for small drops (Tokay

et al., 2013). As rainfall intensity or radar reflectivity are not strongly affected by the concentrations of small drops, no further adjustment of the PSD is considered in this study. For the reconstruction of the drizzle mode of the PSD, Raupach et al. (2019) present a method being able to correct for this deficiency and to further improve rainfall intensity estimates for light rain.

Regarding the measurement of snow, we show that the number of particles with diameters exceeding 0.75 mm is slightly underestimated by the Thies disdrometer. However, this bias is not reflected in intensity estimates. Although not systematically

biased, uncertainty in snowfall intensity estimates is higher than for rainfall as some of the underlying assumptions (e.g. about particle orientation, shape or density) are less appropriate for solid than for liquid precipitation (Yuter et al., 2006; Battaglia et al., 2010). Regarding snow density, we propose a simple parametrization of particle density as a function of particle diameter, which is based on a comparison of aggregated snow volumes and corresponding masses measured by the 2DVD and a weighing precipitation gauge, respectively (1-min observations). The proposed parametrization is similar to other studies (e.g., Fabry and





Szyrmer, 1999; Brandes et al., 2007) and could substantially improve intensity estimates as compared to a constant density assumption. By applying the proposed adjustment of the PSD and the parametrization of snow density, correlation with respect to the OTT pluviometer can be slightly improved. We further tested the inclusion of information about particle velocity and temperature but could not thereby improve resulting intensity estimates.

The distinction between rainfall and snowfall is not only an important prerequisite for the proposed correction methods, but also relevant with respect to hydrological applications in alpine or pre-alpine areas. In this study, we show that the Thies disdrometer is well suited for a distinction of rainfall from snowfall (using the 2DVD as a reference), but that larger differences exist for mixed precipitation and particularly small precipitation intensities. As such, our results are in line with other studies in which precipitation phase estimates from disdrometers (including the Thies disdrometer) have been compared to human

observations (Bloemink and Lanzinger, 2005; Merenti-Välimäki et al., 2001). In particular the underestimation of mixed phased precipitation by the Thies disdrometer is consistent with results of Bloemink and Lanzinger (2005). At the same time, a recently reported case study suggests that the instrument is able to accurately signal mixed precipitation during changes between snow and rain (Pickering et al., 2019). In this context, we would like to point out that the agreement during mixed precipitation with any reference observation will depend on the mixing ratios, which are explicitly or implicitly considered as mixed. In this study,

we presented a particle-by-particle classification algorithm being able to explicitly determine mixing ratios for the reference instrument. Such procedures can provide a basis for the validation of explicitly characterized hydrometeor mixtures as for example in polarimetric radar observations (e.g., Besic et al., 2018) or atmospheric models (e.g., Forbes et al., 2014). However, when comparing dominant hydrometeor type or precipitation phase during a certain time interval, the choice of thresholds is required and will affect the classification and the resulting comparison.

## 5   Conclusions

This study evaluated the Thies disdrometer with respect to precipitation monitoring in a pre-alpine environment; a weighing precipitation gauge (OTT pluviometer) as well as a 2DVD have been used as a reference. We show that the instrument is well suited for precipitation detection. However, in our case, rainfall intensity is systematically underestimated by 16.5%, which may be explained by an underestimation of raindrops with diameters between 0.5 and 3.5 mm. Moreover, we hypothesize that

the general underestimation at our measurement location depends mainly on the prevailing wind conditions and may differ at other measurement sites. In the case of snowfall, no systematic deviations could be found, but uncertainty is generally higher than for rainfall, which might be related to uncertainties with respect to particle orientation, shapes and densities. To slightly improve intensity estimates during snowfall events, we propose to apply an adjustment of the PSD and recalculate intensities by assuming a relationship between particle diameter and density. This relationship was established by combining measurements

of the OTT pluviometer and the 2DVD. Finally, we show that the Thies disdrometer and the 2DVD agree well with respect to the distinction between rain and snowfall (hit rates reaching 99.7 and 95%, respectively) and that therefore an important prerequisite for the proposed correction is fulfilled.

The Thies disdrometer has the advantage of low maintenance requirements and allows not only the estimation of precipitation intensity but also precipitation type. The reliable distinction between rainfall and snowfall is considered here as an advantage for hydrological applications in mountainous environments, where local estimates of precipitation phase are still uncertain. We therefore see a potential in installing disdrometers at sensitive elevations in mountainous catchments complementary to

precipitation gauge and weather radar data to improve precipitation monitoring and short-term flood forecasting in these areas.

The 2DVD was particularly useful in this study to further investigate the biases of the Thies disdrometer, to establish a parametrization of snow density and to provide a reference for the estimation of precipitation phase. Future studies may focus on a refinement of the proposed snow density parametrization and hydrometeor classification algorithm by taking other parameters such as particle orientation or shape (e.g. roundness, oblateness) into account.

In this study, we could not clarify how the relevant underestimation for liquid precipitation is depending on wind or other influences. We suggest to investigate this dependence in further studies.

*Data availability.*  The data used in this study is available on request from Michael Fehlmann (Michael.Fehlmann@unige.ch).

*Author contributions.*  MF designed the framework of the study and evaluation strategy with help from all coauthors. MF processed the data obtained by the measurement devices (i.e. the Thies disdrometer, the two-dimensional video disdrometer, the OTT pluviometer and the

LUFFT weather sensor) and implemented the presented evaluation methods. MF, MR, AvL and MS participated in writing and editing the manuscript, as well as investigating and interpreting the results.

*Competing interests.*  The authors declare that they have no conflict of interest.

*Acknowledgements.*  This work has been supported by the European Horizon 2020 research project ANYWHERE (EC-HORIZON2020-PR700099-ANYWHERE). The authors acknowledge the Canton of Bern for the purchasing of measurement devices, the Federal Office for

the Environment for financial support and Geopraevent for technical support in the field. Also, we would like to thank Joanneum Research and Thies Clima for providing information about data processing algorithms of their instruments (i.e. the Thies disdrometer and the two-dimensional video disdrometer) as well as for the valuable discussion of the obtained results.



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





**Table 1.** Reclassification scheme used for the comparison of dominant precipitation phase (1 min) between the Thies disdrometer and the two-dimensional video disdrometer (2DVD). (Note: Codes in square brackets refer to precipitation types which are not yet identifiable automatically, i.e. these codes are not reported by the instrument.)

| Analysed | Thies disdrometer (SYNOP table 4680) | 2DVD (Equations 1–5) |
|---|---|---|
| Rain | 51,52,53,[54,55,56],57,58,61,62,63,[64,65,66] | Rain >70% |
| Mixed | 67,68 | Rain ≤70% and Snow/ Melting snow/ Graupel ≤80% and Hail ≤1% |
| Snow/ Graupel | 71,72,73,74,75,76,77,[78] | Snow/ Melting snow/ Graupel >80% |
| Hail | 89 | Hail >1% |





**Table 2.** Minimum precipitation thresholds established in the calibration period to optimize precipitation detection for different integration times. The thresholds are chosen to minimise the distance to an ideal point in a receiver operating characteristic (ROC) diagram (i.e. a hit rate equal to 1 and a false alarm rate equal to 0, Fig. 4). The corresponding reduction in hit and false alarm rates as well as the resulting distance to this point are given for the independent validation period.

| $\Delta t$ (min) | Threshold (mm/h) | $\Delta$ Hit rate (%) | $\Delta$ False alarm rate (%) | Distance to optimum (%) |
|---|---|---|---|---|
| 5 | 0.10 | 0 | 0 | 5.2 |
| 10 | 0.05 | 0 | 0 | 5.5 |
| 20 | 0.05 | 2.4 | 2.9 | 5.0 |
| 30 | 0.04 | 2.1 | 3.1 | 4.9 |
| 60 | 0.03 | 2.7 | 4.8 | 5.0 |
| 90 | 0.02 | 1.5 | 5.3 | 5.1 |
| 120 | 0.02 | 2.2 | 6.9 | 4.3 |
| 240 | 0.01 | 2.2 | 8.9 | 5.2 |





**Table 3.** Correction factors for the number of particles in 22 diameter classes as measured by the Thies disdrometer, resulting from a comparison to the two-dimensional video disdrometer in the calibration period. Measurements are separated into rain and snow based on the recorded dominant precipitation type by the Thies disdrometer (1 min).

| Class (no.) | Range (mm) | Rain correction factor | Snow correction factor |
|---:|---|---:|---:|
| 1 | 0.125–0.25 | 0.92 | 0.43 |
| 2 | 0.25–0.375 | 0.83 | 0.55 |
| 3 | 0.375–0.5 | 0.92 | 0.68 |
| 4 | 0.5–0.75 | 1.08 | 0.78 |
| 5 | 0.75–1 | 1.57 | 1.26 |
| 6 | 1–1.25 | 1.40 | 1.25 |
| 7 | 1.25–1.5 | 1.43 | 1.21 |
| 8 | 1.5–1.75 | 1.38 | 1.20 |
| 9 | 1.75–2 | 1.35 | 1.19 |
| 10 | 2–2.5 | 1.29 | 1.20 |
| 11 | 2.5–3 | 1.18 | 1.21 |
| 12 | 3–3.5 | 1.10 | 1.18 |
| 13 | 3.5–4 | 0.92 | 1.12 |
| 14 | 4–4.5 | 0.88 | 1.11 |
| 15 | 4.5–5 | 0.81 | 1.08 |
| 16 | 5–5.5 | 0.63 | 1.04 |
| 17 | 5.5–6 | 0.34 | 1.04 |
| 18 | 6–6.5 | 0.70 | 1.02 |
| 19 | 6.5–7 | 0.29 | 1.02 |
| 20 | 7–7.5 | 0.17 | 1.02 |
| 21 | 7.5–8 | 0.17 | 1.01 |
| 22 | >8 | 0.19 | 1.06 |



**Table 4.** Evaluation of precipitation intensities measured by the Thies disdrometer during the second year of measurement using the OTT pluviometer as a reference. Bias (B) und correlation coefficient (Corr) of the uncorrected measurements are given for all events as well as for rain and snowfall separately (left value). Furthermore, the effect of the two proposed correction methods is shown, i.e. the adjustment to the OTT pluviometer (middle value) and the two-dimensional video disdrometer (right value). The correction methods are thereby established during the first year of measurements.

| Δt (min) | All events | | Rain | | Snow | |
|---|---|---|---|---|---|---|
| | Bias (mm/h) | Corr (-) | Bias (mm/h) | Corr (-) | Bias (mm/h) | Corr (-) |
| 5 | -0.02/<0.01/0.01 | 0.97/0.97/0.97 | -0.23/0.02/0.23 | 0.97/0.97/0.97 | -0.05/-0.09/-0.19 | 0.86/0.86/0.84 |
| 10 | -0.02/<0.01/0.01 | 0.98/0.98/0.98 | -0.19/0.01/0.20 | 0.99/0.99/0.98 | -0.05/-0.08/-0.15 | 0.90/0.90/0.89 |
| 20 | -0.02/-0.01/0.01 | 0.99/0.99/0.98 | -0.16/<0.01/0.16 | 0.99/0.99/0.99 | -0.03/-0.06/-0.11 | 0.92/0.92/0.92 |
| 30 | -0.02/-0.01/0.01 | 0.99/0.99/0.99 | -0.14/<0.01/0.14 | >0.99/>0.99/0.99 | -0.03/-0.05/-0.08 | 0.94/0.94/0.95 |
| 60 | -0.02/-0.01/0.01 | 0.99/0.99/0.99 | -0.11/<0.01/0.11 | >0.99/>0.99/>0.99 | -0.02/-0.04/-0.05 | 0.94/0.94/0.96 |
| 90 | -0.02/-0.01/0.01 | 0.99/0.99/0.99 | -0.10/<0.01/0.10 | >0.99/>0.99/>0.99 | -0.02/-0.04/-0.05 | 0.94/0.94/0.96 |
| 120 | -0.02/-0.01/0.01 | 0.99/0.99/0.99 | -0.08/<0.01/0.09 | >0.99/>0.99/>0.99 | -0.02/-0.03/-0.04 | 0.94/0.94/0.96 |
| 240 | -0.02/-0.01/0.01 | 0.99/0.99/0.99 | -0.06/<0.01/0.07 | >0.99/>0.99/>0.99 | -0.02/-0.03/-0.03 | 0.95/0.95/0.97 |





**Table 5.** Comparison of the precipitation phase detected by the Thies disdrometer (rows) and the two-dimensional video disdrometer (columns). The numbers are given as percentages of the total number of 1 min observations during two years of measurements, which are equal to 2,533 h of precipitation.

|  | Rain | Mixed | Snow | Total |
|---|---|---|---|---|
| Rain | 52.4 | 2.9 | 1.8 | 57.1 |
| Mixed | 0.1 | 0.7 | 0.2 | 1.0 |
| Snow | 0.1 | 0.7 | 41.1 | 41.8 |
| Total | 52.6 | 4.3 | 43.1 | 100.0 |
| Hit rate (%) | 99.7 | 16.6 | 95.3 | 94.2 |
| False alarm rate (%) | 9.9 | 0.3 | 1.3 | 2.9 |





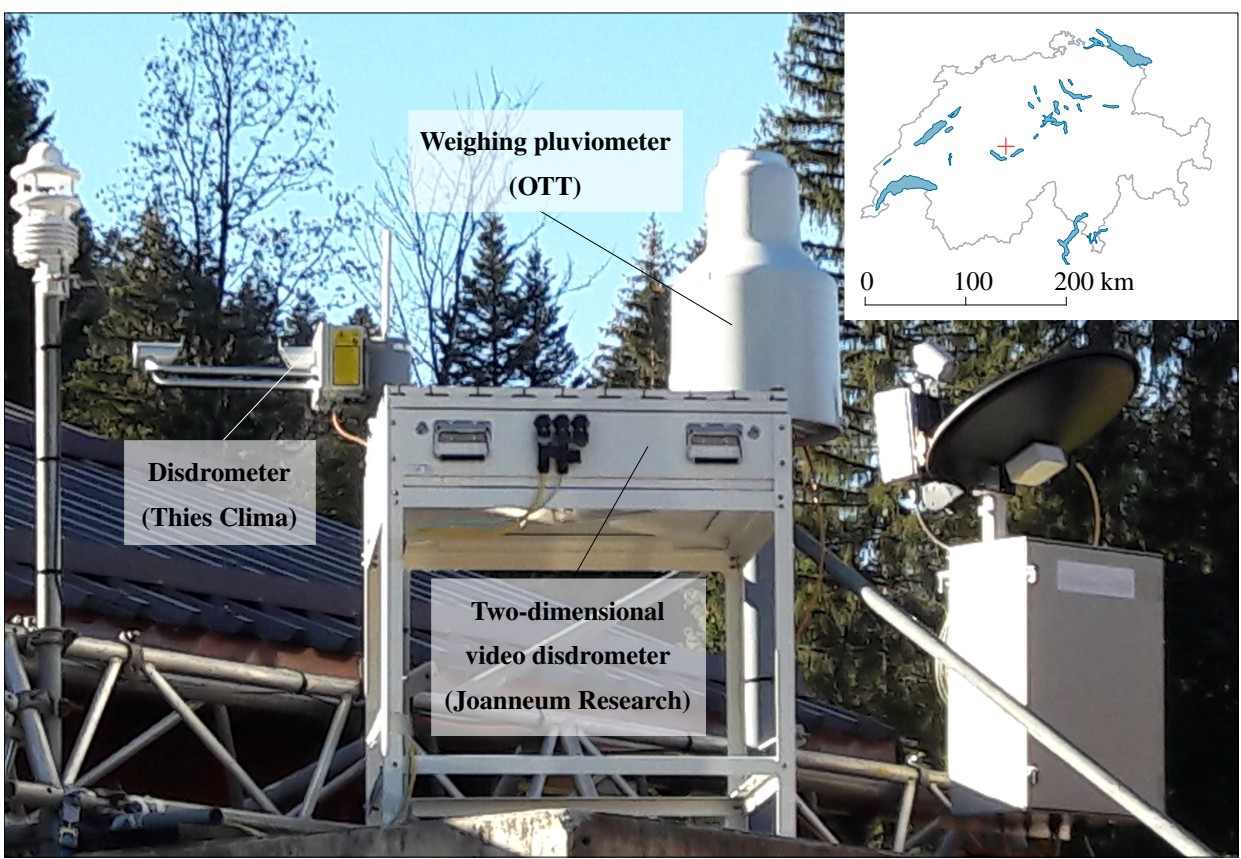

**Figure 1.** Measurement devices located in a pre-alpine area in Switzerland (Innereriz, 1060 m a.s.l.). In this study, the Thies disdrometer is evaluated using both the OTT pluviometer as well as a two-dimensional video disdrometer as a reference during two years of measurements.





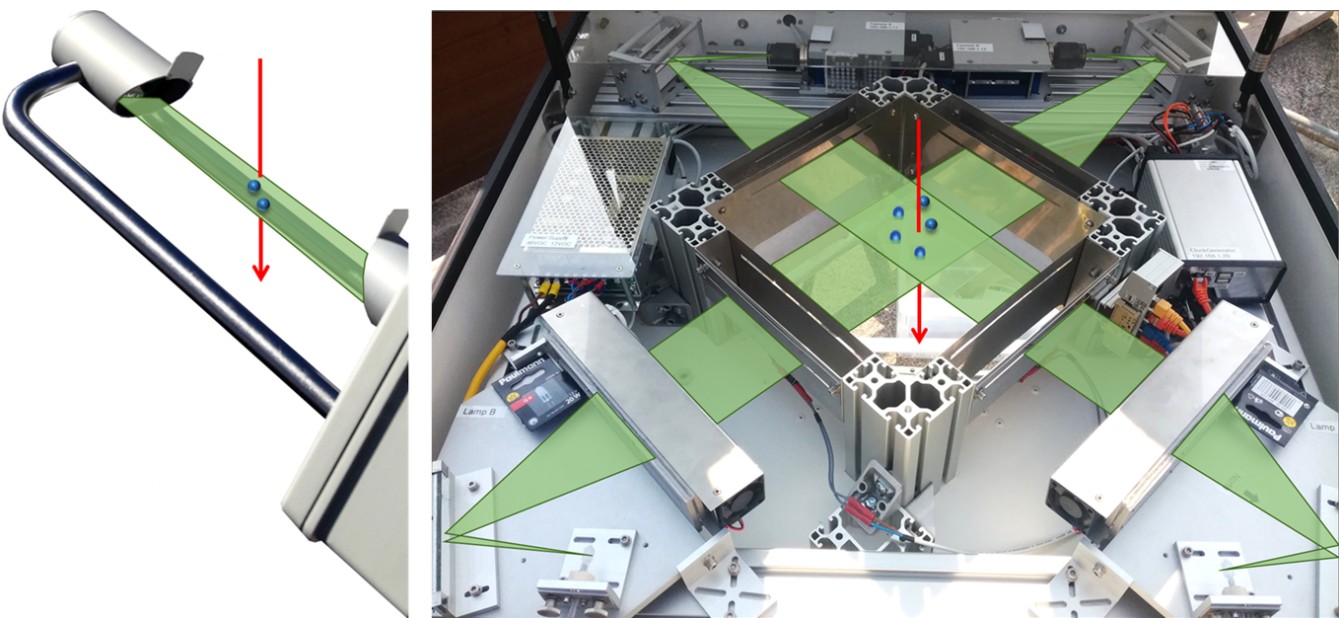

**Figure 2.** Comparison of the measurement principles of the Thies disdrometer (left) and the two-dimensional video disdrometer (2DVD, right): While the Thies disdrometer measures the attenuation of an infrared laser beam (786 nm) by falling particles, the 2DVD detects the shadowing of individual pixels by such particles in images taken by two optical cameras and from two perspectives.



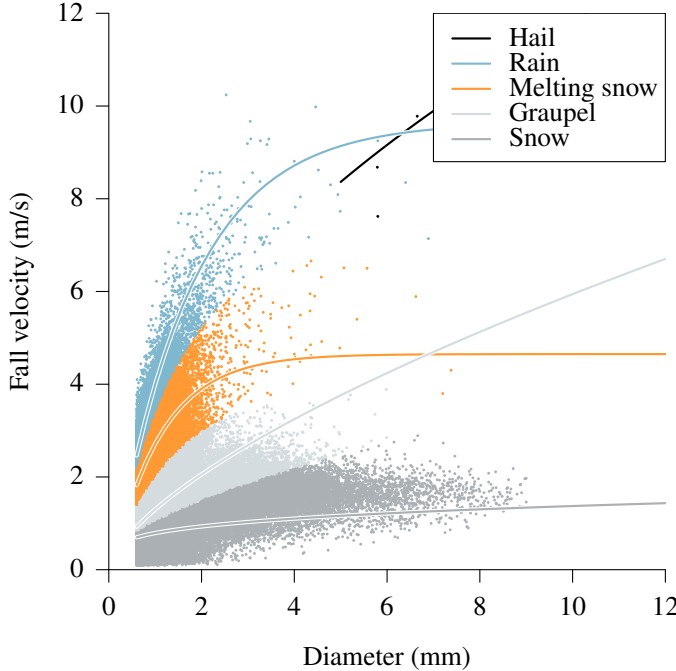

**Figure 3.** Example of the classification algorithm developed in this study during a transition from rain to snowafall (2018-02-17 17:00 to 23:00 UTC). After a plausibility check, each hydrometeor detected by the two-dimensional video disdrometer is classified as one of 5 precipitation types (hail, rain, melting snow, graupel, snow). This classification is based on empirical relationships between particle diameter and fall velocity (equations 1–5).





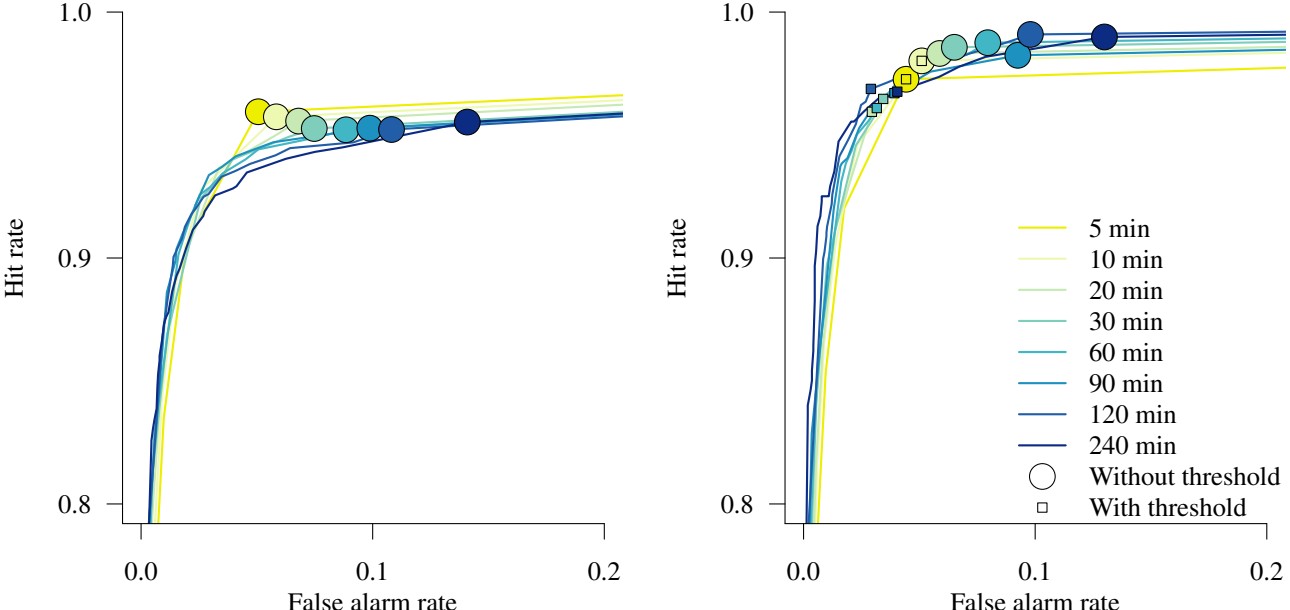

**Figure 4.** Receiver operating characteristic (ROC) curves showing hit and false alarm rates of the Thies disdrometer with respect to the detection of precipitation using the OTT pluviometer as a reference. Left: Exploration of hit and false alarm rates during the whole time series (two years). Right: Effect on applying minimum precipitation thresholds on hit and false alarm rates during the second year of measurements. Note that the proposed thresholds are established during the first year of measurements in order to reduce false alarm rates, particularly for longer integration times.





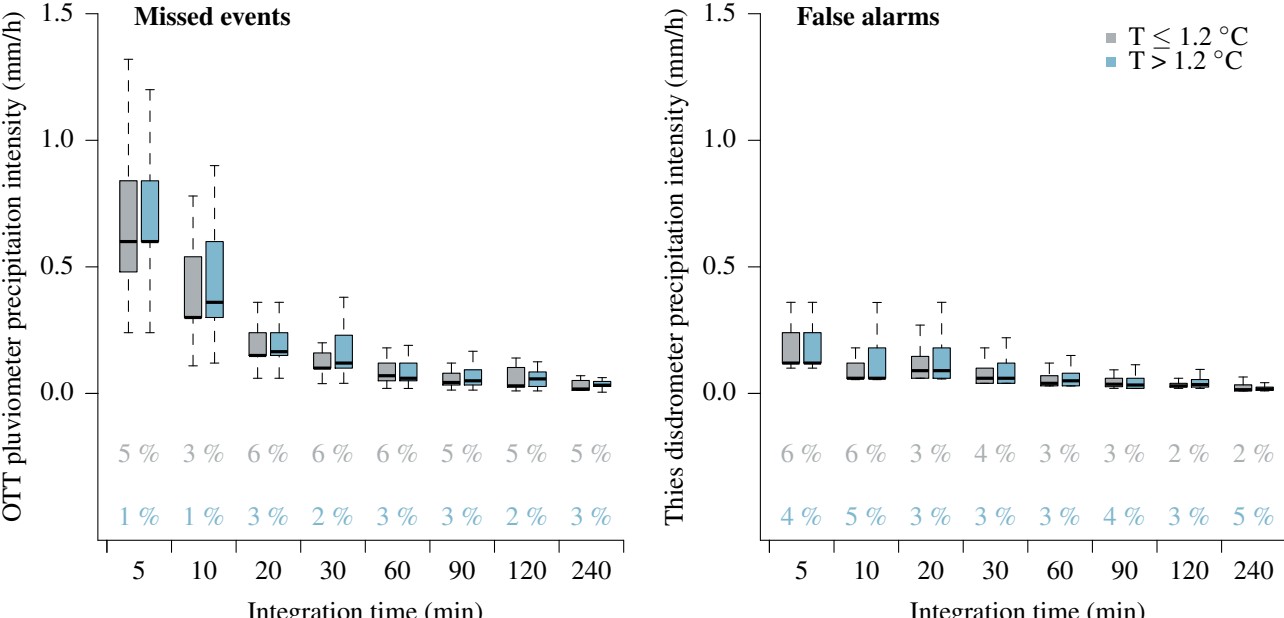

**Figure 5.** Distribution of precipitation intensities and phase during missed events (left) and false alarms (right) by the Thies disdrometer during the validation period (each box shows the median and interquartile range of the distribution while the whiskers extend to 1.5 times this range from the box or to the most extreme data point). While precipitation intensities measured by the OTT pluviometer are analysed during missed events, precipitation intensities indicated by the Thies disdrometer are analysed during false alarms. Events are separated according to a temperature threshold (1.2 °C), and the relative frequency of missed events as well as the false alarm rate is given at the bottom of each panel for cases above and below this temperature threshold.





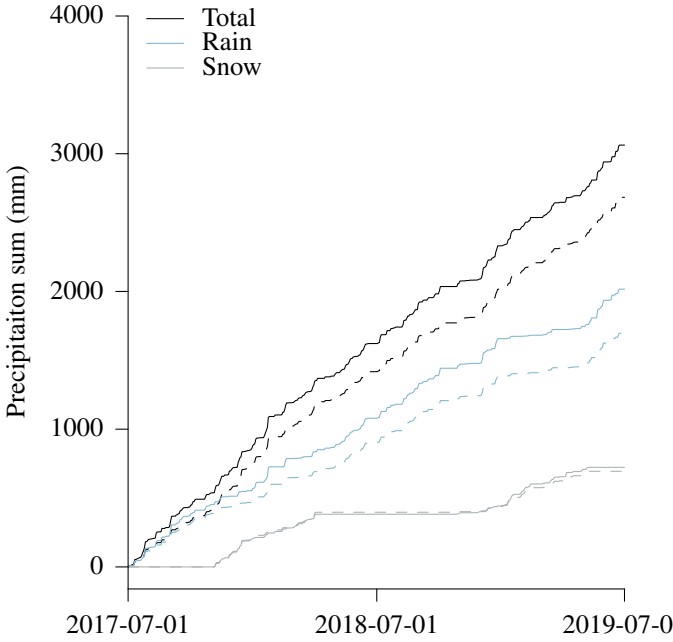

**Figure 6.** Cumulative precipitation sums as measured by the Thies disdrometer (dashed lines) and the OTT pluviometer (solid lines) during the whole time series (two years). Precipitation sums are separated into rain and snow based on the recorded dominant precipitation type by the Thies disdrometer (1 min).





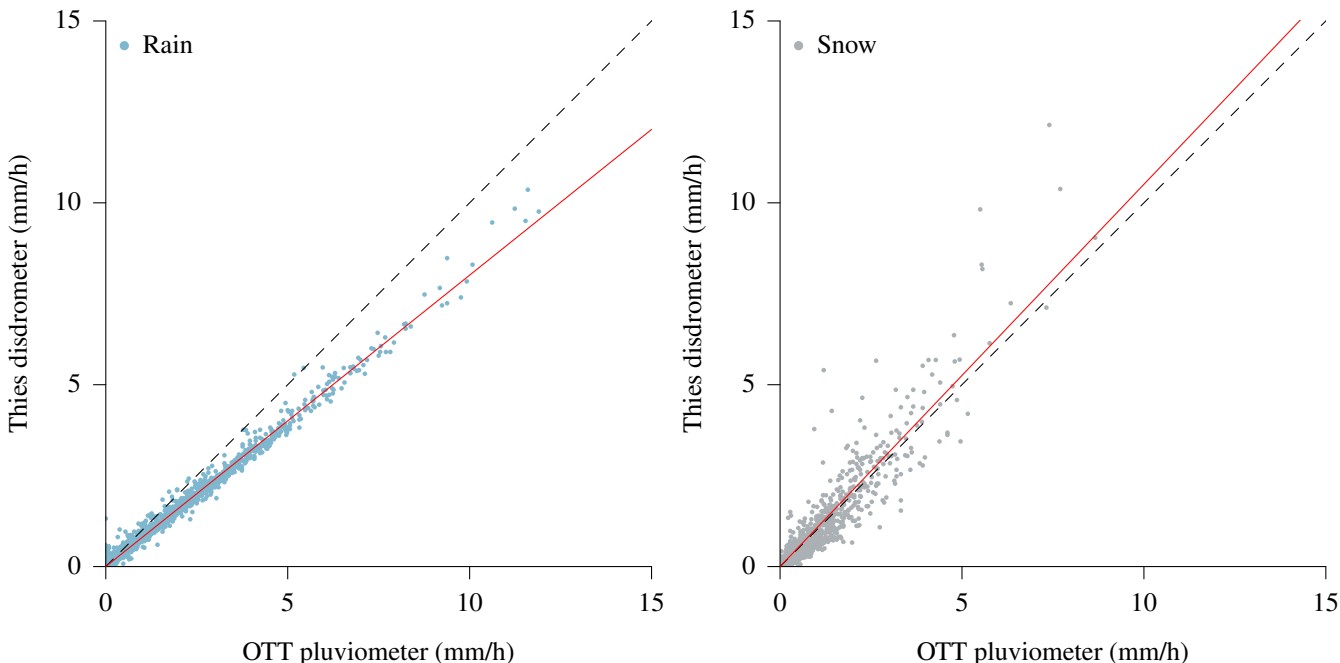

**Figure 7.** Precipitation intensities during periods of 30 min as recorded by the Thies disdrometer and the OTT pluviometer during the whole time series (two years). Precipitation is separated into rain (left) and snow (right) based on the recorded dominant precipitation type by the Thies disdrometer (1 min). To highlight systematic errors, a linear regression is shown in both panels (red line), which is forced through the origin and has a slope of 0.80 for rainfall intensities and of 1.05 for snowfall intensities.





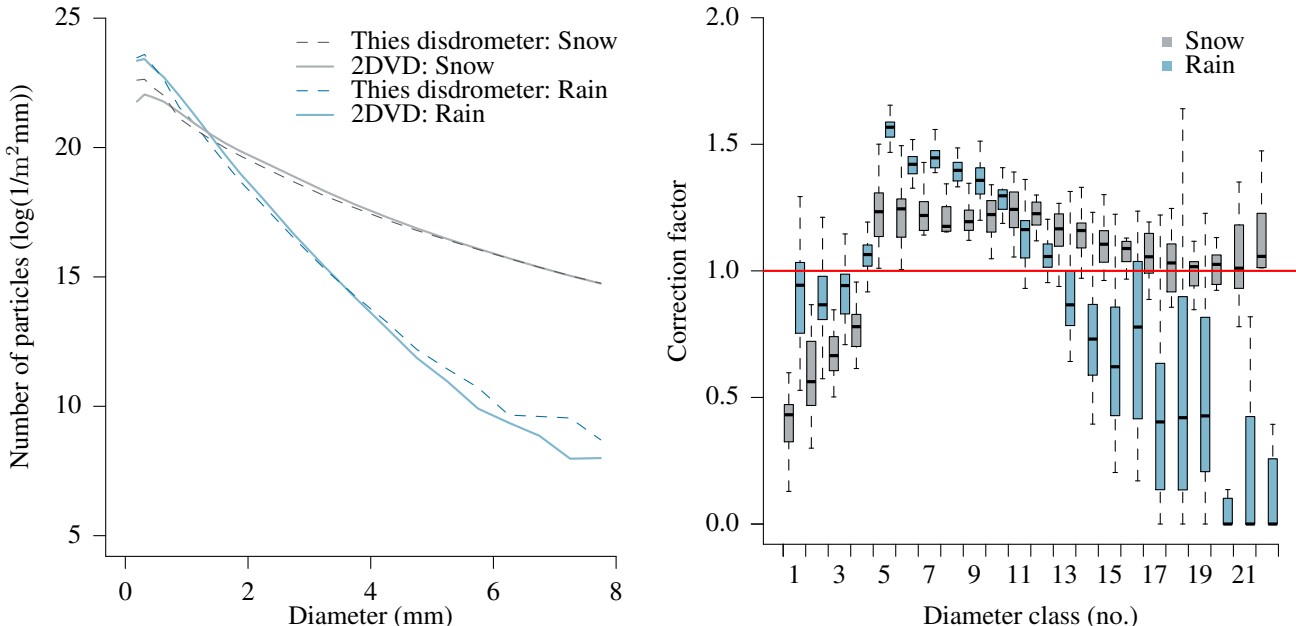

**Figure 8.** Comparison of particle size distribution (PSD) obtained by the Thies disdrometer and the two-dimensional video disdrometer (2DVD) during the whole time series (two years). Left: Summed PSD during all observed rain and snowfall events. The separation into rain and snowfall events is based on the recorded dominant precipitation type by the Thies disdrometer (1 min). Right: Resulting correction factors for different diameter classes of the Thies disdrometer, using the 2DVD as a reference. Thereby, the median and variability of these correction factors is shown using monthly results (each box shows the interquartile range of the distribution while the whiskers extend to 1.5 times this range from the box or to the most extreme data point).





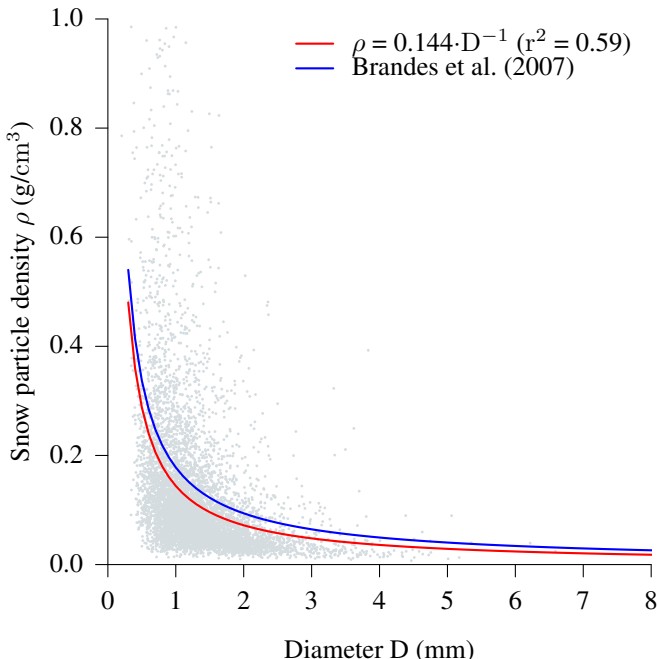

**Figure 9.** Relationship between snow particle density and mean particle diameter based on 1 min observations during the first year of measurements. Snowfall events are identified based on the recorded dominant precipitation type by the Thies disdrometer. Snow particle density is then calculated by comparing the precipitation volume measured by the two-dimensional video disdrometer (2DVD) and precipitation mass measured by the OTT pluviometer, and related to mean particle diameter as measured by the 2DVD. The fitted curve is used to translate particle size distribution into snowfall intensities during the second year of measurements. Note: The corresponding relationship established by Brandes et al. (2007) is shown as a reference.





**Figure 10.** Precipitation intensities during periods of 30 min as recorded by the Thies disdrometer and the OTT pluviometer during the second year of measurements). Precipitation is separated into rain, snow and other types (e.g. mixed) based on the recorded dominant precipitation type by the Thies disdrometer (1 min observation). The effect of the two proposed correction methods, i.e. adjustment to the OTT pluviometer and the two-dimensional video disdrometer (2DVD), are shown in separate panels (B: Bias; Corr: Correlation coefficient). Note that these adjustments distinguish between rain and snowfall and were established in the first year of measurements.





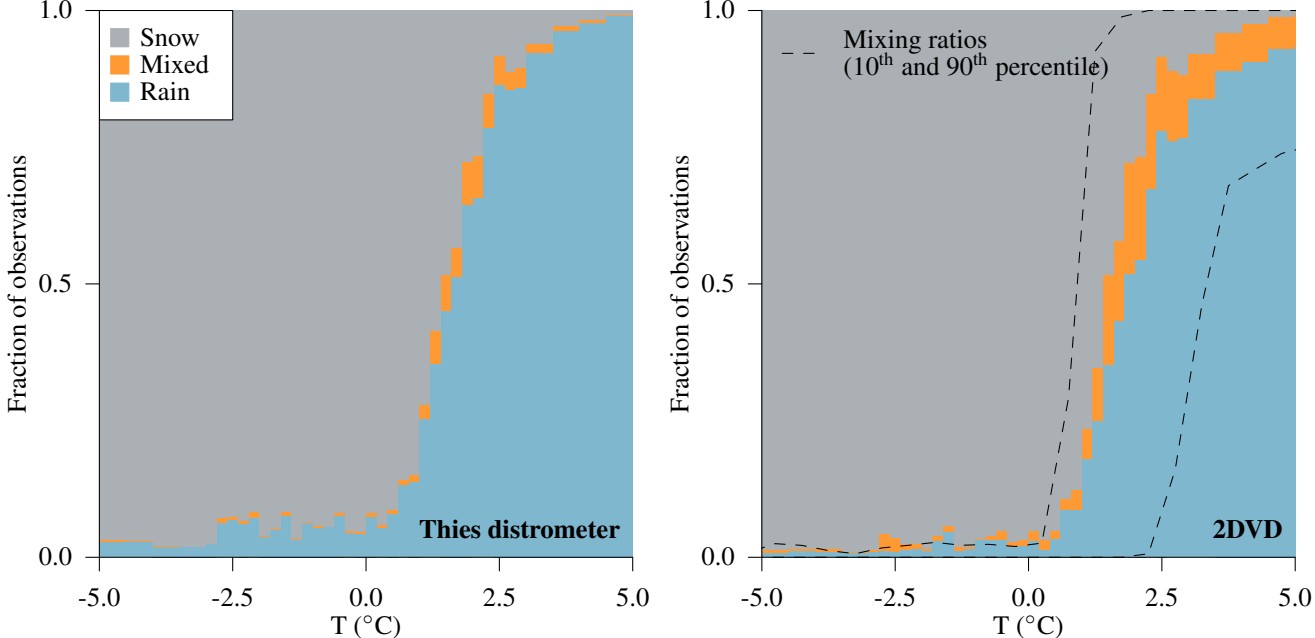

**Figure 11.** Relative frequency of the observed dominant precipitation phase by the Thies disdrometer (left) and the two-dimensional video disdrometer (2DVD, right) as a function of air temperature during two years of measurements (2,533 h of precipitation).