# Peer review of "Automated precipitation monitoring with the Thies disdrometer: Biases and ways for improvement"

_Atmospheric Measurement Techniques, 2019_

## Short Comment (SC1) · 7 Feb 2020

Dear authors,

Thanks for making your work open to comments. I might be able provide a couple of useful comments hereby, having worked with Thies LPM Clima instruments recently. I find your paper very well written and organised, and easy to follow. A couple of comments below are listed as dot points in no particular order of importance:

1. I find that your introduction might benefit from adding further explanations, in particular when it comes to the use of laser disdrometers outside of precipitation amounts measurements; e.g. gathering of DSD for parameterisation of models and retrievals. Typically line 20, you mention the "verification of dual-pol radars" but it is not reduced

to this, and you could possibly mention all the different usage of the DSD (not only for the Thies) but for disdrometers in general.

2. Lines 21 and 22: "not many studies have assessed uncertainties of disdrometers" – this is not really correct, and quite perilous to state this without including a succinct literature review. There are plenty of studies assessing uncertainties of disdrometers (usually by comparing disdrometers of different make-up / manufacturers / principles, or/and co-located instruments), but they often investigate the OTT Parsivels (both versions) and 2DVD in their majority. For the Thies in particular, you could mention here Angulo-Martinez et al. (2018) and Guyot et al. (2019), both published in the companion EGU-journal HESS.

Angulo-Martínez, M., Beguería, S., Latorre, B., & Fernández-Raga, M.: Comparison of precipitation measurements by OTT Parsivel 2 and Thies LPM optical disdrometers. Hydrology and Earth System Sciences, 22(5), 2811, https://doi.org/10.5194/hess-22-2811-2018, 2018.

Guyot, A., Pudashine, J., Protat, A., Uijlenhoet, R., Pauwels, V. R. N., Seed, A., and Walker, J. P.: Effect of disdrometer type on rain drop size distribution characterisation: a new dataset for south-eastern Australia, Hydrol. Earth Syst. Sci., 23, 4737–4761, https://doi.org/10.5194/hess-23-4737-2019, 2019.

In these two studies, measurements of rainfall are evaluated using respectively OTT Parsivel1 and 2 and Thies LPM. This could serve as well for your discussion, in particular when it comes to the uncertainties and systematic under-estimation of rainfall by Thies instruments. We find in Guyot et al. (2019) that Thies underestimated liquid precipitation when compared to the OTT Parsivels (1 and 2).

3. Line 28 to 30. I believe these findings have been revisited in Thurai et al. (2016) and later Thurai and Bringi (2018), Raupach et al. (2019)? The 2DVD seems to underestimate droplets in the lower range of diameters (< 0.5 mm), meaning that their use as reference can be questionable in some circumstances in particular over that range.

Overall, it would be great to mention that there is no perfect reference that one can use, and each instrument will be affected by uncertainties. For the 2DVD, it would be great to mention that the literature is evolving and previous findings might not hold anymore or only partially.

Thurai, M. and Bringi, V. N.: Application of the generalized gamma model to represent the full rain drop size distribution spectra, J. Appl. Meteorol. Clim., 57, 1197–1210, https://doi.org/10.1175/jamc-d-17-0235.1, 2018.

Thurai, M., Gatlin, P., Bringi, V. N., Petersen, W., Kennedy, P., Notaroš, B., & Carey, L. (2017). Toward completing the raindrop size spectrum: Case studies involving 2D-video disdrometer, droplet spectrometer, and polarimetric radar measurements. Journal of Applied Meteorology and Climatology, 56(4), 877-896.

Raupach, T. H., Thurai, M., Bringi, V. N., & Berne, A. (2019). Reconstructing the drizzle mode of the raindrop size distribution using double-moment normalization. Journal of Applied Meteorology and Climatology, 58(1), 145-164.

4. In your manuscript, it would be great to differentiate the two types of Parsivel (1 and 2) using a superscript, as in the second version; the manufacturer has corrected some issues in particular in the lower range of diameters.

5. In terms of rainfall, we have found in Guyot et al. (2019) that the Thies starts to underestimate the number of droplets from 0.75 mm onwards towards larger diameters (instead of 0.5 mm as mentioned in your paper) when compared to Parsivel1. Since we do not use the same reference (in your case 2DVD), this might explain the difference but again here I think it is good to keep in mind that 2DVD is not an absolute reference and has been questioned for his accuracy in the recent literature.

6. Data availability: It adds a great value to the work to make the data accessible openly on a repository (and possibly the code as well, mentioning libraries having been used if any to give credits to the authors). One of the strengths of open-access articles is

also to promote that accessibility of data and code so that work can be re-produced, and data shared easily.

Thanks for making it possible to read and comment on your work, I enjoyed the reading.

With kind regards,

Adrien Guyot Monash University, Australia
* * *

---

## Referee Comment (RC1) · Anonymous Referee #1 · 18 Feb 2020

**REVIEW REPORT**

Review of amt-2019-466

By Michael Fehlmann, Mario Rohrer, Annakaisa von Lerber, and Markus Stoffel

Manuscript Title – Automated precipitation monitoring with the Thies disdrometer: Biases and ways for improvement

**GENERAL COMMENTS**

The manuscript describe the performance of Thies Clima disdrometer with respect to OTT pluviometer and 2D video disdrometer in terms of precipitation detection, precipitation amount and intensity and classification between rain, snow and mix phase. Furthermore the manuscript describe two methodology to correct the Thies data and analyzed the effects of these methods on the precipitation intensity. The paper is well written and organized. Following there are some specific comments. I suggest the publication on AMT after addressing my comments.

**SPECIFIC COMMENTS**

- Page 2. Lines 20-25: Several studies have been done to evaluate the performance of Thies Clima and some references related to this topic need to be added in the Introduction section. At that regard, following there are some suggestions
    - Lanza et al. 2012 and Lanzinger et al. (2006) described the result of a a WMO experiment that showed a bias that range between 5% and 20% comparing rain gauge and Thies Clima disdrometer rainfall amount
    - In Upton et al. (2008), Angulo-Martínez et al.(2017), and Adirosi et al. (2018), the Thies Clima perfromance has been evaluated with respect to Parsivel disdrometer.

- Section 2: Did the Authors applied any filtering method to eliminate the so called "spurious drops" due to win, splashing, or mismatch? Several studies that used disdrometer measured DSD applied a filter criterion based on fall velocity such as the one adopted in Tokay et al. 2001 and valid only for rain.
- Section 2: Different classification methods are applied to Thies disdrometer and 2DVD data to distinguish between rain, snow and mixed phase. Is it possible to applied the proposed classification method to Thies data (of course applying the method to binned data instead of drop-by-drop data)? In this way the obtained results can be compared with the classification provided by the Thies software. If not, why do not apply a classification method that can be easily applied to 2DVD and Thies data? It can help to exclude the possible effect of the application of different classification methodologies on the obtained results
- Section 2: Is there a minimum values of precipitation amount that can be detected by OTT pluviometer? Such as the 0.2 mm for the tipping bucket gauge?
- Page 6 last sentence: it is not clear to me. Please clarify.
- Page 7, first paragraph: which is the range of variability of the thresholds used to obtain the ROC diagram? The threshold are applied to both disdrometer and gauge data?
- Figure 8 right and Table 3: How do the Authors compute the correction factor in these cases?

- Page 9, third line: "This suggest…….intensities". Looking at the results obtain for rainy minutes in terms of bias it seems that the adjustment to the OTT pluviometer is the one that reduces the bias while the other adjustment provides same or higher bias values. In all the other columns of Table 4 the differences between the uncorrected data, the data corrected with OTT pluviometer and the data corrected with 2DVD are  negligible! Please provide a more detailed comment on this

MINOR COMMENT

- Figure 3: please move the legend. In this position it covers the data
- Figure 6: Check x-label

REFERENCES

Adirosi, E., Roberto, N., Montopoli, M., Gorgucci, E., & Baldini, L. (2018). Influence of disdrometer type on weather radar algorithms from measured DSD: Application to Italian climatology. Atmosphere, 9(9), 360.

Angulo-Martínez, M.; Beguería, S.; Latorre, B.; Fernández-Raga, M. Comparison of precipitation measurements by Ott Parsivel2 and Thies LPM optical disdrometers. Hydrol. Earth Syst. Sci. Discuss. 2017.

Lanza, L.G.; Vuerich, E. Non-parametric analysis of one-minute rain intensity measurements from the WMO Field Intercomparison. Atmos. Res. 2012, 103, 52–59.

Lanzinger, E.; Theel, M.; Windolph, H. Rainfall amount and intensity measured by the Thies laser precipitation monitor. In Proceedings of the WMO Technical Conference on Meteorological and Environmental Instruments and Methods of Observation (TECO), Geneva, Switzerland, 4–6 December 2006

Tokay, A., Kruger, A., & Krajewski, W. F. (2001). Comparison of drop size distribution measurements by impact and optical disdrometers. Journal of Applied Meteorology, 40(11), 2083-2097.

Upton, G.; Brawn, D. An investigation of factors affecting the accuracy of thies disdrometers. In Proceedings of the Technical Conference on Instruments and Methods of Observation (TECO-2008), St. Petersburg, Russia, 27–28 November 2008.

---

## Referee Comment (RC2) · Anonymous Referee #2 · 18 Mar 2020

Review of amt-2019-466

**Manuscript title**: Automated precipitation monitoring with the Thies disdrometer: Biases and ways for improvement

**Authors:** Michael Fehlmann, Mario Rohrer, Annakaisa von Lerber, and Markus Stoffel

The manuscript describes the capabilities of Thies disdrometer in both quantifying the amount and identifying the type of precipitation. To this end, OTT pluviometer and 2DVD are used as reference, respectively. The results show an underestimation of the precipitation amount, while a good capabilities in identifying the precipitation type. The analysis about the precipitation detection in terms of categorical scores (i.e. hits, false alarm, miss, etc.) has to be improved since it is not too clear and a number of questions arise: in particular, the analysis about ROC and the use or not of a minimum precipitation threshold. On the other hand, the comparison with 2DVD is clearer and useful.

The paper is useful since it shows how much reliable is the Thies disdrometer in measuring the precipitation, but before to be accepted for publication the authors have to address the following comments.

− Figure 2 has to be improved. The size of 2DVD picture may be reduced.

− Page 3, line 27: please, modify the sentence because the 2DVD is not based on a similar principle than Thies.

− Page 4, lines 29-30: it is not clear if the Eq. 1-5 are applied in the Thies precipitation classification algorithm or if different relationship are applied. Please, clarify this.

− Page 6, lines 16-20: in my opinion, the correlation coefficient is not one of the best indicators for this type of analysis (the Table 4 confirm this, showing high CC values before and after the Thies correction). Figure 7 shows that the data are distributed along a straight line, but this is not close to the one-to-one line (as should be). A more indicative indicator to associate to the bias could be the root mean square error.

− Page 6, lines 27-28: what does it mean "…with respect to precipitation detection…"? Is it a minimum precipitation threshold or what? And, is it referred to the OTT pluviometer or to the Thies? It is almost impossible to understand by reading the text.

− Page 7, lines 1-3: by looking at Figure 4, the combination of hits and false alarm can exceed or not 100%. Obviously, when the sum is lower than 100% it is because of miss and/or correct negative, but what about when the sum exceed 100? Is it always because they are calculated with respect to precipitation detection? This reviewer (and this could be true for a reader) is not familiar with ROC, but the text should allow to understand the methodology.

− Page 7, lines 8-9: I am always skeptic when an instrument like a disdrometer or pluviometer is considered to be able to detected so weak precipitation.

− Figure 5: a logarithmic scale on the y-axis could be better.

– Page 8, line 8: or "…described above. Whereas…" or "…described above: whereas…"

– Page 8, line 11: the mean ratio of what?

– Page 8, lines 15-16: the PSD shown in Figure 8 are obtained by averaging all the 1-minute PSDs collected during the two years?

– Page 8, lines 34-35: I basically agree that the impact of both correction methods are comparable, but the "2DVD correction" gives higher bias than "OTT pluviometer correction". This could indicate a slight overestimation of the precipitation by 2DVD if compared to the OTT pluviometer.

– Page 9, lines 13-14: the sample size information should not be reported here but at the beginning of Section 2.3.

– Page 10, lines 18-21: to state that the correction method proposed by you and the one proposed in Raupach and Berne (2015) are consistent you should apply their method to your data (only because Thies and OTT Parsivel are based on the principle).

– Conclusions: the first part pf the Conclusions (i.e. page 11, lines 21-32) is a summary of Section 4. I suggest merging the two sections in only one that could be titled "Discussion and Colclusions".

---

## Author Comment (AC1) · 17 Jun 2020

Specific comments

1.1 Page 2. Lines 20-25: Several studies have been done to evaluate the performance of Thies Clima and some references related to this topic need to be added in the Introduction section. At that regard, following there are some suggestions - Lanza et al. 2012 and Lanzinger et al. (2006) described the result of a a WMO experiment that showed a bias that range between 5% and 20% comparing rain gauge and Thies Clima disdrometer rainfall amount - In Upton et al. (2008), Angulo-Martínez et al.(2017), and Adirosi et al. (2018), the Thies Clima perfromance has been evaluated with respect to Parsivel disdrometer.

[Figure]

Response: Thank you very much for this comment and for suggesting additional literature. Also in accordance with a short comment (SC), we have added the suggested literature and reformulated this part of the introduction (please also see corresponding SC).

Changes in the manuscript (section 1): However, there are only few studies which assess the uncertainties of the Thies disdrometer, mostly comparing the instrument to OTT Parsivel disdrometers (e.g., Adirosi et al., 2018; Angulo-Martínez et al., 2018; Guyot et al., 2019; Upton and Brawn, 2008) and in a few cases to rain gauges (e.g., Lanza and Vuerich, 2012; Lanzinger et al., 2006).

1.2 Section 2: Did the Authors applied any filtering method to eliminate the so called "spurious drops" due to win, splashing, or mismatch? Several studies that used disdrometer measured DSD applied a filter criterion based on fall velocity such as the one adopted in Tokay et al. 2001 and valid only for rain.

Response: Thank you very much for this comment. Indeed, this effect exists and several studies apply filter algorithms to remove spurious measurements of mostly larger particles from 2DVD or other video disdrometer measurements by applying a filter based on the combined velocity-diameter information (e.g., von Lerber et al., 2017; Raupach and Berne, 2015). However, as shown by Friedrich et al. (2013) such effects mostly occur at high wind speeds (exceeding 20 m/s) and as our study is extremely wind sheltered we did not see the need of applying such a filter in this study. This is described now in the revised manuscript.

Changes in the manuscript (section 2.1): Note that in some studies using 2DVD or other video disdrometer measurements, additional filters are applied to remove spurious measurements of mostly larger particles, usually being based on a validity check of the combined diameter and velocity information (e.g., Raupach and Berne, 2015; von Lerber et al., 2017). However, as investigated in detail by Friedrich et al. (2013) such spurious measurements mostly occur at wind speeds exceeding 20 m/s. As our

study site is extremely wind sheltered (see also section 4) we thus did not apply such a filter in this study.

1.3 Section 2: Different classification methods are applied to Thies disdrometer and 2DVD data to distinguish between rain, snow and mixed phase. Is it possible to applied the proposed classification method to Thies data (of course applying the method to binned data instead of drop-by-drop data)? In this way the obtained results can be compared with the classification provided by the Thies software. If not, why do not apply a classification method that can be easily applied to 2DVD and Thies data? It can help to exclude the possible effect of the application of different classification methodologies on the obtained results

Response: Thank you very much for this comment and proposal of this additional analysis. It is indeed possible to apply the proposed classification method to the Thies data, given the limitation that only binned data instead of drop-by-drop data is available as you mention correctly.

As you proposed, we have applied the classification method to the binned Thies data, using the "centroid" (i.e. the mean velocity V and mean diameter D) of each V-D class to assign a precipitation type to all particles within the corresponding V-D class. The result is shown in Fig. 1 and 2 in analogy to Table 5 and Fig. 11 in the manuscript.

As can be seen in Fig. 1 compared to Fig. 11 in the manuscript, the proposed classification method results in a very similar classification of snow when applied to binned Thies disdrometer data and 2DVD data, respectivly. The hit rate of the Thies disdrometer with respect to the 2DVD is even slightly higher than for the classification of the Thies software (98.0% compared to 95.3% in Table 5 of the manuscript). For liquid and mixed precipitation, higher differences exist: when applying the proposed classification method to the binned Thies data, more "mixed"precipitation is detected at temperatures > ~1°C than when applying the same classification to 2DVD data. The hit rate of the Thies disdrometer with respect to the 2DVD for liquid precipitation therefore also drops

to 86% and at the same time the hit rate for mixed precipitation increases to 60.2% (as compared to 99.7% and 16.6% in Table 5 of the manuscript, i.e. when using the Thies classification software).

The most interesting result in our opinion is here that we find that the classification of liquid vs. "mixed" precipitation is very sensitive to the choice of the thresholds used for the assignment of "mixed" precipitation. In our opinion, the problem of consistently defining "mixed" precipitation already exists for human observations, but will be more pronounced when replacing human with automated observation. We also have included the following statement in the original manuscript: "In this context, we would like to point out that the agreement during mixed precipitation with any reference observation will depend on the mixing ratios, which are explicitly or implicitly considered as mixed."

The additional analysis presented here indicates that any reasonable definition of mixing ratios considered as "mixed" precipitation will furthermore depend on the instrument used, which we state now also in the revised manuscript.

Changes in the manuscript (sections 2.2 and 4): 2.2: To investigate the effect of applying different classification methodologies on obtained results, the classification algorithm described above was also applied to Thies data. Given the binned data, the mean velocity and diameter of each V-D class were used for the classification rather than information about individual particles. 4: Our analysis indicates that the distinction between liquid and mixed precipitation is particularly sensitive to the choice of such a threshold. Furthermore, the application of the proposed classification algorithm to both disdrometers indicates that a reasonable choice of these thresholds might differ between different instruments.

1.4 Section 2: Is there a minimum values of precipitation amount that can be detected by OTT pluviometer? Such as the 0.2 mm for the tipping bucket gauge? Response: Thank you for this comment. We added the sensitivity according to the manual of the

[Figure]

OTT pluviometer as well as a description of insights from a study applying a minimum threshold for weak precipitation to this instrument in the description of the measurement devices (2.1). Regarding a careful interpretation of results, we have already covered this aspect in the manuscript in our opinion and refer to the corresponding response of reviewer #2 for further comments. Changes in the manuscript (section 2.1): According to the operating instructions of the OTT pluviometer, the instrument provides the raw precipitation values every 6 seconds using a resolution of 0.001 mm. After the application of special filter algorithms (e.g. a correction for wind effects), non-real time 1-min outputs are available at a resolution of 0.01 mm. Of course, it can be questioned whether very weak precipitation can actually be measured so accurately. For example, Tiira et al. (2016) found in their mass retrieval (performed approximately every 5 minutes) that the output seems to fluctuate and used a threshold 0.2 mm/h for their analysis. 1.5 Page 6 last sentence: it is not clear to me. Please clarify. Response: Thank you for this comment. The sentence was indeed not very clear. Also, it is rather an interpretation than a description of the result, so we moved this statement to the discussion (section 4) and try to explain the meaning/ interpretation of the false alarm rate in more detail there. This is also in accordance with a comment of reviewer # 2 (please check the corresponding comment for more details).

To understand our interpretation please consider the following: - Given is a number of corresponding Thies and OTT pluviometer observations in terms of precipitation yes/ no. - Using the OTT pluviometer as a reference, the false alarm rate of the Thies disdrometer is defined as: FAR = # false alarms / (# false alarms + # correct negatives),

or in other words: FAR = # of cases where Thies = precip and OTT pluviometer = no precip / # of cases where OTT pluviometer = no precip.

To make this even more intuitive, we can interpret the FAR as follows: Given a period without precipitation according to the reference instrument (OTT pluviometer), the FAR can be interpreted as the probability of the evaluated instrument (Thies disdrometer) nevertheless indicating precipitation during this period. - Observed behaviour in Figure

4 (left): The FAR of the Thies disdrometer increases with increasing length of the observations considered, i.e.: given a very long 'dry period' (= no percip) in the reference instrument, the probability of the Thies disdrometer to indicate precipitation is higher than for a very short 'dry period'. - Interpretation: This behaviour can be expected when assuming that the Thies disdrometer is wrongly indicating precipitation at more or less regular time intervals. In this case, "the chance of misinterpreting a signal as precipitation is increasing with increasing integration time" as stated in the text. This could either be due to a regularly occurring misinterpretation of a signal/ disturbance as precipitation or indeed be related to very weak precipitation events which are not detected by the OTT pluviometer. This critical view on the reference instrument (OTT pluviometer) is also emphasized by reviewer # 2 and has led to some changes in the revised manuscript (please check the corresponding comment for more details).

Changes in the manuscript (sections 3.1 and 4): The statement was removed from the results (section 3.1). Instead, we added the following to the discussion (section 4): The false alarm rate, which indicates the probability of the Thies disdrometer detecting precipitation during a dry period, is increasing with increasing integration time. This can be somewhat expected, as the chance of misinterpreting a signal or disturbance as precipitation is increasing with increasing duration of this period. Furthermore, false alarm rates might be affected by the sensitivity of the reference instrument, but are comparable to findings of Bloemink. and Lanzinger (2005) who use human observations as a reference.

1.6 Page 7, first paragraph: which is the range of variability of the thresholds used to obtain the ROC diagram? The threshold are applied to both disdrometer and gauge data?

Response: Thank you for this comment. We used fixed thresholds THROC in mm/h for all integration times with THROC = {0, 0.001, 0.002, . . ., 0.05, 0.1, 0.15, . . ., 1, 1.2, 1.4, . . ., 3}. The thresholds are only applied to the measurements of the Thies disdrometer, as the OTT pluviometer is used as a reference instrument representing

the "ground truth". This was indeed not clarified in the text and was added in the revised manuscript together with a more detailed description of the concept of ROC curves (see comment 2.6 of reviewer #2).

Changes in the manuscript (section 2.3): In the case of precipitation detection (yes/ no), we further investigate the effect of minimum precipitation thresholds applied to measurements of the Thies disdrometer on hit and false alarm rates by investigating the so-called Receiver Operating Characteristic (ROC) curves (e.g., Jolliffe and Stephenson, 2012). A ROC curve thereby depicts the variation of hit and false alarm rates with the variation of such a threshold. For example, using a threshold of 0 mm/h for precipitation detection (i.e. always reporting precipitation regardless of the measurement) will result in both a hit and a false alarm rate of 1. On the other hand, choosing an indefinitely high minimum precipitation threshold will result in both a hit and a false alarm rate of 0. Between these extremes, the resulting hit and false alarm rates depend on the capabilities of the Thies disdrometer to detect precipitation as compared to the reference instrument, while the theoretical optimum (hit rate of 1 and false alarm rate of 0) can usually not be achieved. To establish ROC curves for different integration times we use the fixed thresholds THROC = {0, 0.001, 0.002, . . ., 0.05, 0.1, 0.15, . . ., 1, 1.2, 1.4, . . ., 3} in mm/h.

1.7 Figure 8 right and Table 3: How do the Authors compute the correction factor in these cases?

Response: Thank you for this comment. We follow the method proposed by Raupach and Berne (2015) simply using the ratio of drop concentrations per diameter class as correction factors. While Raupach and Berne (2015) use the median ratio over multiple time periods, we use the ratio of the summed drop concentration over the whole calibration period. The description can be found in section 2.3 (p. 6 line 10ff in the original manuscript) and was slightly extended in the revised manuscript.

Changes in the manuscript (section 2.3): The correction factors used for this scaling

correspond to the ratio of summed 2DVD drop concentrations to summed Thies drop concentrations in the calibration period (2017-07-01–2018-06-30), and are separately calculated for rain and snowfall.

1.8 Page 9, third line: "This suggest. . . . . ..intensities". Looking at the results obtain for rainy minutes in terms of bias it seems that the adjustment to the OTT pluviometer is the one that reduces the bias while the other adjustment provides same or higher bias values. In all the other columns of Table 4 the differences between the uncorrected data, the data corrected with OTT pluviometer and the data corrected with 2DVD are negligible! Please provide a more detailed comment on this

Response: Thank you very much for this valuable comment. This was also brought up by reviwer # 2 and we have indeed not interpreted Table 4 detailed enough. As you state correctly, the adjustment to the OTT pluviometer is able to reduce the bias for liquid precipitation also in the validation period whereas the adjustment to the 2DVD introduced a positive bias. With regard to snowfall, both correction methods have only a small impact and even slightly increase the bias. With regard to correlation, the linear adjustment has no effect by nature, while the adjustment to the 2DVD can slightly improve correlation with respect to snowfall.

Based on these observations, we would clearly recommend to use the proposed adjustment to the OTT pluviometer for correcting the estimation of liquid precipitation intensities and state this in the revised manuscript. If interested in the drop spectra the adjustment to the 2DVD could nevertheless be of interest. Also, the analysis of the PSD is seen as valuable in this study to investigate possible reasons of the biases in precipitation intensity estimates, which we state in the conclusions (section 4).

Changes in the manuscript: (please see answer to corresponding comment of reviewer # 2 for more information on changes in sections 3.2 and 4 in the revised manuscript.)

Minor comments

1.10 Figure 3: please move the legend. In this position it covers the data

Response: Thank you for this comment. We have moved the legend accordingly in the revised manuscript.

Changes in the manuscript (Figure 3): The legend is plotted outside the plot window.

1.11 Figure 6: Check x-label

Response: Thank you very much for this comment and observation. We changed the x-label so that the full date (2019-07-01) is displayed in the revised manuscript.

Changes in the manuscript (Figure 6): Change in x-label so that the full date (2019-07-01) is displayed

References Adirosi, E., Roberto, N., Montopoli, M., Gorgucci, E., & Baldini, L. (2018). Influence of disdrometer type on weather radar algorithms from measured DSD: Application to Italian climatology. Atmosphere, 9(9), 360.

Angulo-Martínez, M.; Beguería, S.; Latorre, B.; Fernández-Raga, M. Comparison of precipitation measurements by Ott Parsivel2 and Thies LPM optical disdrometers. Hydrol. Earth Syst. Sci. Discuss. 2017. Lanza, L.G.; Vuerich, E. Non-parametric analysis of one-minute rain intensity measurements from the WMO Field Intercomparison. Atmos. Res. 2012, 103, 52–59.

Lanzinger, E.; Theel, M.; Windolph, H. Rainfall amount and intensity measured by the Thies laser precipitation monitor. In Proceedings of the WMO Technical Conference on Meteorological and Environmental Instruments and Methods of Observation (TECO), Geneva, Switzerland, 4–6 December 2006

Tokay, A., Kruger, A., & Krajewski, W. F. (2001). Comparison of drop size distribution measurements by impact and optical disdrometers. Journal of Applied Meteorology, 40(11), 2083-2097.

Upton, G.; Brawn, D. An investigation of factors affecting the accuracy of thies disdrometers. In Proceedings of the Technical Conference on Instruments and Methods of Observation (TECO-2008), St. Petersburg, Russia, 27–28 November 2008.

[Figure]

**Fig. 1.** Relative frequency of the observed dominant precipitation phase by the classification algorithm applied to binned Thies disdrometer data as a function of air temperature during two years of measuremen

|                         | Rain | Mixed | Snow  | Total |
|-------------------------|------|-------|-------|-------|
| Rain                    | 45.2 | 1.4   | 0.1   | 46.7  |
| Mixed                   | 6.8  | 2.6   | 0.8   | 10.1  |
| Snow                    | 0.6  | 0.3   | 42.3  | 43.2  |
| Total                   | 52.6 | 4.3   | 43.2  | 100.0 |
| Hit rate (\\%)          | 86.0 | 60.2  | 98.0  | NA    |
| False alarm rate (\\%)  | 3.1  | 7.9   | 1.6   | NA    |

**Fig. 2.** Comparison of the precipitation phase detected by the Thies disdrometer (rows) and the two-dimensional video disdrometer (columns), applying the classification algorithm proposed in section 2.2 to bot

---

## Author Comment (AC2) · 17 Jun 2020

Specific comments

2.1 Figure 2 has to be improved. The size of 2DVD picture may be reduced.

Response: Thank you for this comment. I reduced the size of the compound image to 0.7 of its original width and also tried to visually separate the two subfigures. I am not sure in what other way I should improve the figure in your opinion. Please bring this up again if you had other changes in mind.

Changes in manuscript (Figure 2): Figure reduced to 0.7 of its original width.

2.2 Page 3, line 27: please, modify the sentence because the 2DVD is not based on a

similar principle than Thies.

Response: Thank you for this comment. We removed this statement in the corresponding sentence of the revised manuscript.

Changes in the manuscript (section 2.1): The 2DVD, developed by Joanneum research, [statement removed] is able to derive more direct and more detailed information about individual hydrometeors than the Thies disdrometer.

2.3 Page 4, lines 29-30: it is not clear if the Eq. 1-5 are applied in the Thies precipitation classification algorithm or if different relationship are applied. Please, clarify this.

Response: Thank you very much for this comment, this was indeed not stated clearly enough. Unfortunately, the exact empirical relations used in the Thies precipitation classification algorithm (except Gunn and Kinzer, 1949) are not reported by the manufacturer. We were also not able to get more insight into other details of their algorithm. We made this now more explicit in both the description of the Thies distrometer (2.1) as well as in the description of the classification algorithm used to process the 2DVD data (2.2). Following a suggestion of reviewer #1 on this topic, we also applied our classification method to the raw data of the Thies disdrometer and mention implications of this analysis in the discussion of the revised manuscript (please see corresponding comment of reviewer #1).

Changes in the manuscript (sections 2.1 and 2.2): 2.1: The exact functioning of this classification algorithm as well as other equations used are thereby not reported by the manufacturer. 2.2: To investigate the effect of applying different classification methodologies on obtained results, the classification algorithm described above was also applied to Thies data. Given the binned data, the mean velocity and diameter of each V-D class were used for the classification rather than information about individual particles.

2.4 Page 6, lines 16-20: in my opinion, the correlation coefficient is not one of the best indicators for this type of analysis (the Table 4 confirm this, showing high CC values

before and after the Thies correction). Figure 7 shows that the data are distributed along a straight line, but this is not close to the one-to-one line (as should be). A more indicative indicator to associate to the bias could be the root mean square error.

Response: Thank you very much for this comment. The advantage of the CC compared to many other metrics (including the RMSE) is that it is independent of any bias, i.e. reflects the scatter between the two observations independent of any systematic deviations. Our original intention was to provide the CC as a metric of how this scatter can be reduced by an adjustment of the Thies PSD to the 2DVD. However, as you point out correctly, the CC only changes very little before and after the adjustment to the 2DVD, i.e. the correction is mainly affecting the bias. Also, in case of the linear adjustment to the OTT pluviometer, the CC remains by nature unaffected. As the paper is actually focusing on biases and the added value of including the CC is limited, we agreed to remove the CC from Table 4 and we agree that this helps to keep the paper more focused. However, we still mention in the text that a slight improvement of the CC can be achieved with respect to snowfall intensities when using the adjustment to the 2DVD.

Regarding the characterisation of the bias, we would however like to stick to the used metric of the absolute bias for the following reasons. The advantage of this metric is that it is unaffected by the scatter and furthermore can be easily interpreted by the reader. The RMSE, on the other hand, can increase with both increasing bias or increasing scatter, and is a more complex measure probably less intuitive for the reader to interpret.

Changes in the manuscript (Table 4, sections 2.3, 3.2 and 4, Equation 7): The correlation coefficient in Table 4 and related descriptions in section 3.2 are removed. Also, the description of the methodology in section 2.3 is shortened and equation 7 is removed. Note that we keep the statement related to the improvement of snowfall intensity estimates in the discussion (please see 3rd paragraph of section 4 in revised manuscript).

2.5 Page 6, lines 27-28: what does it mean "…with respect to precipitation detection…"? Is it a minimum precipitation threshold or what? And, is it referred to the OTT pluviometer or to the Thies? It is almost impossible to understand by reading the text.

Response: Thank you for this comment. It means that we have investigated the capability of the Thies disdrometer to distinguish precipitation from no precipitation (binary variable). The hit and false alarm rates are given for the Thies disdrometer as stated in this sentence. The OTT pluviometer is used as a reference (i.e. representing the 'ground truth'), which is stated multiple times in the manuscript, e.g. in the first sentence of 3.1, page 6, line 23. The hit and false alarm rates described in this sentence refer to the comparison without introducing thresholds. The effect of applying minimum precipitation thresholds is described in the following paragraph (page 7, lines 1ff.). I have tried to reformulate the sentence in the manuscript.

Changes in the manuscript (section 3.1): The capability of the Thies disdrometer to distinguish precipitation from no precipitation is described in terms of its hit and false alarm rate when using the OTT pluviometer as a reference. In a first step, hit and false alarm rates are calculated over the whole time series and are indicated with circles in Fig. 4 (left) for different integration times. […] In a second step, we tested the application of minimum precipitation thresholds to the Thies disdrometer observations in order to reduce false alarm rates for longer integration times.

2.6 Page 7, lines 1-3: by looking at Figure 4, the combination of hits and false alarm can exceed or not 100%. Obviously, when the sum is lower than 100% it is because of miss and/or correct negative, but what about when the sum exceed 100? Is it always because they are calculated with respect to precipitation detection? This reviewer (and this could be true for a reader) is not familiar with ROC, but the text should allow to understand the methodology.

Response: Thank you for this comment. You are right that we have not explained the

concept of a ROC curve sufficiently in the original manuscript and we added a more detailed description in the revised manuscript, also following a more technical comment (1.6) of reviwer #1. Regarding the concept of hit and false alarm rates, we would like to keep the reference to Jolliffe and Stephenson (2012) as we think with the given example a reader will get the correct understanding of these concepts.

To clarify your specific question: yes, the sum of hit and false alarm rate can exceed 1, as they both can take values from 0 to 1 independent from each other. Considering the following example: an overly sensitive measurement device which always reports precipitation will achieve a hit every time there is actually precipitation and no misses. The hit rate = hits/(hits+misses) will be 1. However, the same instrument will always produce a false alarm every time there is actually no precipitation and no correct negatives. The false alarm rate = false alarms/(false alarms+correct negatives) will also be 1. Thus, the sum of hit rate and false alarm rate will be 2. When imagining the opposite, i.e. a totally insensitive instrument which never reports precipitation, it will be clear that both the hit rate and the false alarm rate will be 0. In reality, the combination lies somewhere between these extremes, depending on the capabilities of the instrument, and can be further changed (ex-post) by introducing a minimum precipitation threshold for the instrument. The theoretical optimum (hit rate of 1 and false alarm rate of 0), however, can usually not be achieved.

Changes in the manuscript (section 2.3): For the evaluation of categorical variables, i.e. precipitation detection (yes/ no) and precipitation phase (rain/ mixed/ snow), hit and false alarm rates with respect to the reference instrument are calculated according to Jolliffe and Stephenson (2012). In the case of precipitation detection (yes/ no), we further investigate the effect of minimum precipitation thresholds applied to measurements of the Thies disdrometer on hit and false alarm rates by investigating the so-called Receiver Operating Characteristic (ROC) curves (e.g., Jolliffe and Stephenson, 2012). A ROC curve thereby depicts the variation of hit and false alarm rates with the variation of such a threshold. For example, using a threshold of 0 mm/h for precipitation detection (i.e. always reporting precipitation regardless of the measurement) will result in both a hit and a false alarm rate of 1. On the other hand, choosing an indefinitely high minimum precipitation threshold will result in both a hit and a false alarm rate of 0. Between these extremes, the resulting hit and false alarm rates depend on the capabilities of the Thies disdrometer to detect precipitation as compared to the reference instrument, while the theoretical optimum (hit rate of 1 and false alarm rate of 0) can usually not be achieved. To establish ROC curves for different integration times we use the fixed thresholds THROC = {0, 0.001, 0.002, . . ., 0.05, 0.1, 0.15, . . ., 1, 1.2, 1.4, . . ., 3} in mm/h.

2.7 Page 7, lines 8-9: I am always skeptic when an instrument like a disdrometer or pluviometer is considered to be able to detected so weak precipitation.

Response: Thank you very much for this comment. Of course, we are also sceptic towards the capability of these instruments to detect so weak precipitation – although according to the user manuals, the OTT pluviometer can detect precipitation > 0.01 mm and the Thies disdrometer even provides minimal intensities of 0.001 mm/h for drizzle.

With the analysis provided in the manuscript we are nevertheless able to show that the two instruments agree quite well with respect to precipitation detection. Also, we can show that – when using the OTT pluviometer as a reference – an even better agreement is achieved with the introduction of minimum precipitation thresholds for the Thies disdrometer. Given the difficulties of measuring so weak precipitation, we agree, however, that it is difficult to determine the real 'ground truth' or to make absolute statements about the capabilities of each instrument with respect to this 'ground truth'. That is also the reason, why we state in the discussion that "false alarm rates might be affected by the sensitivity of the reference instrument, but are [at least] comparable to findings of Bloemink and Lanzinger (2005) who use human observations as a reference".

Note that also reviewer # 1 asked to include more information about minimum precipitation amounts detected by the OTT pluviometer, which we included in the revised manuscript.

Changes in the manuscript: (See corresponding comment of reviewer # 1 for more information about minimum precipitation amounts detected by the OTT pluviometer.)

2.8 Figure 5: a logarithmic scale on the y-axis could be better.

Response: Thank you for this comment. Indeed, boxplot ranges for low precipitation intensities can be better read when using a logarithmic scale on the y-axis. We changed the two subfigures accordingly and added a hint to the logarithmic scale in the figure caption.

Changes in the manuscript (Figure 5): A logarithmic scale is used on the y-axis and the following hint is added to the figure caption. "Note that a logarithmic scale is used to display precipitation intensities."

2.9 Page 8, line 8: or "...described above.  Whereas..." or "...described above: whereas..."

Response: Thank you for this comment. We changed the manuscript according to the second suggestion above.

Changes in the manuscript (section 3.2): "...described above: whereas..."

2.10 Page 8, line 11: the mean ratio of what?

Response: Thank you for this comment. We were using the terminology of Raupach and Berne (2015) here, apparently without explicitly stating it. The mean ratio is defined as the reference mean divided by the observed mean, while the 'mean' refers to the mean over a certain number of time steps. Alternatively, this can be expressed as the ratio of the reference to the observed precipitation sum in the calibration period, which is probably somewhat easier to understand for the reader and has been changed accordingly in the revised manuscript.

Changes in the manuscript (section 3.2): We thus propose to use the ratio of the OTT pluviometer to the Thies disdrometer precipitation sum as a correction factor and to distinguish between rain and snowfall. Using the first year of measurements, the resulting correction factors for rain and snowfall intensities are 1.20 and 0.96, respectively.

2.11 Page 8, lines 15-16: the PSD shown in Figure 8 are obtained by averaging all the 1-minute PSDs collected during the two years?

Response: Figure 8 actually shows the distribution of the summed (not averaged) 1-min PSDs over two years, which is explained in the figure caption as follows: "Comparison of particle size distribution (PSD) obtained by the Thies disdrometer and the two-dimensional video disdrometer (2DVD) during the whole time series (two years). Left: Summed PSD during all observed rain and snowfall events. The separation into rain and snowfall events is based on the recorded dominant precipitation type by the Thies disdrometer (1 min)." To make this more explicit also in the text, we revised the corresponding sentence in the text of the revised manuscript.

Changes in the manuscript: A comparison of the summed PSD between these two instruments is shown in Fig. 8 for all rain and snowfall events during the whole time series (two years). The separation into rain and snowfall events is based on the recorded dominant precipitation type by the Thies disdrometer (1 min).

2.12 Page 8, lines 34-35: I basically agree that the impact of both correction methods are comparable, but the "2DVD correction" gives higher bias than "OTT pluviometer correction". This could indicate a slight overestimation of the precipitation by 2DVD if compared to the OTT pluviometer.

Response: Thank you very much for this valuable comment. A similar comment was also made by reviewer #1, and we have indeed not interpreted Table 4 detailed enough. It seems indeed that for liquid precipitation, the adjustment to the 2DVD introduces a positive bias of roughly the same magnitude as the negative bias without adjustment in the validation period. As stated in your comment, this could indicate a slight overestimation of liquid precipitation by the 2DVD when compared to the OTT pluviometer. Therefore, we would clearly recommend to apply the more robust adjustment to the OTT pluviometer. We adjusted the description of Table 4 in the result (section 3.2) as well as the discussion (section 4) accordingly.

Changes in the manuscript (sections 3.2 and 4): 3.2: As can be seen in Table 4 and Fig. 10, the most robust result is achieved by the adjustment of rainfall intensities to the OTT pluviometer, which successfully reduces the underestimation of liquid precipitation in the validation period. The adjustment of rainfall intensities to the 2DVD, however, results in a positive bias in the validation period. For snowfall, both correction methods have a smaller impact and even result even in slightly higher negative biases than are present without any adjustment. 4: To reduce the underestimation of rainfall intensities by the Thies disdrometer, we established an adjustment to 2DVD measurements following the methodology of Raupach and Berne (2015). However, when applying the resulting adjustment in the validation period, we introduce a positive bias, which could indicate a slight overestimation of liquid precipitation by the 2DVD when compared to the OTT pluviometer. A more stable correction is achieved by applying a linear adjustment to the OTT pluviometer. This method is thus proposed as the preferred correction method in this study, especially when the PSD itself is not of interest to the user.

2.13 Page 9, lines 13-14: the sample size information should not be reported here but at the beginning of Section 2.3.

Response: Thank you for this comment. We have moved the corresponding sentence from section 3.3 to 2.3 where the comparison between the two disdrometers with respect to precipitation type detection is described.

Changes in the manuscript (sections 3.3 and 2.3): The following sentence is moved from section 3.3 to 2.3: "Furthermore, we only consider pairwise complete (1 min) observations of both instruments with either rain, snow or mixed precipitation, resulting in a time series of 2,533 h of precipitation."

2.14 Page 10, lines 18-21: to state that the correction method proposed by you and the one proposed in Raupach and Berne (2015) are consistent you should apply their method to your data (only because Thies and OTT Parsivel are based on the principle).

Response: Thank you for this comment. I am not sure if I understand your comment fully, but would like to provide some clarifications before I come to the changes made in the revised manuscript. The application of the exact same correction method as proposed by Raupach and Berne (2015) to our data is not really possible, as they establish their correction to Parsivel disdrometers (generations 1 and 2) and we are evaluating the Thies disdrometer. However, as Raupach and Berne (2015) highlight in their article, the "the correction can be trained on and applied to data from [. . .] any disdrometer in general." So rather than applying their method, we adopt their methodology. This distinction was made more clearly in the revised manuscript.

Furthermore, in the original sentence, we only state that our result is consistent with the result in Raupach and Berne (2015) in so far as the correlation coefficient is only slightly affected by their correction of the Parsivel as well as our correction of the Thies disdrometer. However, as you have expressed yourself critically towards the use of the correlation coefficient in an earlier comment, we have removed this statement from the revised manuscript.

Changes in the manuscript (section 4): (Please see response to earlier comments for corresponding changes in the revised manuscript.)

2.15 Conclusions: the first part pf the Conclusions (i.e. page 11, lines 21-32) is a summary of Section 4. I suggest merging the two sections in only one that could be titled "Discussion and Colclusions".

Response: Thank you for this comment. You are right and we merged the two sections as proposed. We thereby removed lines 21-32, keeping only one statement of it to the new, merged section (see below).

Changes in the manuscript (sections 1 and 4): the Following statement was kept from lines 21-32: "hit rates reaching 99.7% for rainfall and 95% for snowfall using the 2DVD as a reference", the rest was removed. Furthermore, the description of the structure of the paper in the Introduction was changed accordingly: "In section 4, results are discussed and conclusions are drawn with respect to the operational monitoring of precipitation with the Thies disdrometer as well as potential applications in a hydrological context."
* * *

---

## Author Comment (AC3) · 17 Jun 2020

Specific comments

3.1 I find that your introduction might benefit from adding further explanations, in particular when it comes to the use of laser disdrometers outside of precipitation amounts measurements; e.g. gathering of DSD for parameterisation of models and retrievals. Typically line 20, you mention the "verification of dual-pol radars" but it is not reduced to this, and you could possibly mention all the different usage of the DSD (not only for the Thies) but for disdrometers in general.

Response: You are right. We amended further usages.

Changes in the manuscript (section 1): Beside the calibration and verification of rain-

fall estimation by radar and satellite, disdrometers are also used for a proper understanding of hydrometeorological regimes and soil erosion, pollution wash off in urban environments or interactions of rainfall with crop and forest canopies (Angulo-Martinez, 2018; Frasson and Krajewski, 2011; Nanko et al., 2004; Nanko et al., 2013).

3.2 Lines 21 and 22: "not many studies have assessed uncertainties of disdrometers" – this is not really correct, and quite perilous to state this without including a succinct literature review. There are plenty of studies assessing uncertainties of disdrometers (usually by comparing disdrometers of different make-up / manufacturers / principles, or/and co-located instruments), but they often investigate the OTT Parsivels (both versions) and 2DVD in their majority. For the Thies in particular, you could mention here Angulo-Martinez et al. (2018) and Guyot et al. (2019), both published in the companion EGU-journal HESS.

Angulo-Martínez, M., Beguería, S., Latorre, B., & Fernández-Raga, M.: Comparison of precipitation measurements by OTT Parsivel 2 and Thies LPM optical disdrometers. Hydrology and Earth System Sciences, 22(5), 2811, https://doi.org/10.5194/hess-22-2811-2018, 2018.

Guyot, A., Pudashine, J., Protat, A., Uijlenhoet, R., Pauwels, V. R. N., Seed, A., and Walker, J. P.: Effect of disdrometer type on rain drop size distribution characterisation: a new dataset for south-eastern Australia, Hydrol. Earth Syst. Sci., 23, 4737–4761, https://doi.org/10.5194/hess-23-4737-2019, 2019.

In these two studies, measurements of rainfall are evaluated using respectively OTT Parsivel1 and 2 and Thies LPM. This could serve as well for your discussion, in particular when it comes to the uncertainties and systematic under-estimation of rainfall by Thies instruments. We find in Guyot et al. (2019) that Thies underestimated liquid precipitation when compared to the OTT Parsivels (1 and 2).

Response: Thank you, you are right. We changed the sentence mentioning that there are only few studies mentioning the uncertainties of the Thies distrometer, including

also literature suggested by reviewer #1 (please see corresponding comment). Also, we included a sentence in the discussion mentioning Guyot et al. (2019).

Changes in the manuscript (sections 1 and 4): We changed the sentence in the introduction: However, there are only few studies which assess the uncertainties of the Thies disdrometer, mostly comparing the instrument to OTT Parsivel disdrometers (e.g., Adirosi et al., 2018; Angulo-Martínez et al., 2018; Guyot et al., 2019; Upton and Brawn, 2008) and in a few cases to rain gauges (e.g., Lanza and Vuerich, 2012; Lanzinger et al., 2006). In the discussion we added: Finally, when compared the OTT Parsivels, Guyot et al. (2019) found that the Thies disdrometer [. . .] underestimates liquid precipitation compared to both Parsivel1 and Parsivel2.

3.3 Line 28 to 30. I believe these findings have been revisited in Thurai et al. (2016) and later Thurai and Bringi (2018), Raupach et al. (2019)? The 2DVD seems to underestimate droplets in the lower range of diameters (< 0.5 mm), meaning that their use as reference can be questionable in some circumstances in particular over that range. Overall, it would be great to mention that there is no perfect reference that one can use, and each instrument will be affected by uncertainties. For the 2DVD, it would be great to mention that the literature is evolving and previous findings might not hold anymore or only partially.

Thurai, M. and Bringi, V. N.: Application of the generalized gamma model to represent the full rain drop size distribution spectra, J. Appl. Meteorol. Clim., 57, 1197–1210, https://doi.org/10.1175/jamc-d-17-0235.1, 2018.

Thurai, M., Gatlin, P., Bringi, V. N., Petersen, W., Kennedy, P., Notaroš, B., & Carey, L. (2017). Toward completing the raindrop size spectrum: Case studies involving 2Dvideo disdrometer, droplet spectrometer, and polarimetric radar measurements. Journal of Applied Meteorology and Climatology, 56(4), 877-896.

Raupach, T. H., Thurai, M., Bringi, V. N., & Berne, A. (2019). Reconstructing the drizzle mode of the raindrop size distribution using double-moment normalization. Journal of

Applied Meteorology and Climatology, 58(1), 145-164.

Response: We think at the end we have to have some reference, but we added that the 2DVD seems to underestimate small particles

Changes in the manuscript (section 1): We added the following to the introduction: "...,even if the 2DVD seems to underestimate droplets in the lower range of diameters, i.e. below 0.5 mm (Raupach et al., 2019; Thurai et al., 2017; Thurai and Bringi , 2018).

3.4 In your manuscript, it would be great to differentiate the two types of Parsivel (1 and 2) using a superscript, as in the second version; the manufacturer has corrected some issues in particular in the lower range of diameters.

Response: Thank you for this comment, we have tried to better make this distinction when referring explicitly to one of these instruments

Changes in the manuscript (section 4): In addition to other comments added during this revision, this was changed in the following sentence of the manuscript: "For example, the OTT Parsivel1 disdrometer only underestimates drops with sizes ranging between 0.8 and 1.6 mm and only during periods of higher rainfall intensity."

3.5 In terms of rainfall, we have found in Guyot et al. (2019) that the Thies starts to underestimate the number of droplets from 0.75 mm onwards towards larger diameters (instead of 0.5 mm as mentioned in your paper) when compared to Parsivel1. Since we do not use the same reference (in your case 2DVD), this might explain the difference but again here I think it is good to keep in mind that 2DVD is not an absolute reference and has been questioned for his accuracy in the recent literature.

Response: Thank you for this hint, as you are mentioning we are using the 2dvd as a reference. As Raupach and Berne (2015) are writing: If a better reference becomes available, exactly the same approach could be applied to correct the Parsivel (or indeed any other disdrometer) and to improve the agreement with the reference.

Changes in the manuscript (section 4): Guyot et al. (2019) found that the Thies disdrometer starts to underestimate the number of droplets from 0.75 mm onwards towards larger diameters when compared to Parsivel1. . .

3.6 Data availability: It adds a great value to the work to make the data accessible openly on a repository (and possibly the code as well, mentioning libraries having been used if any to give credits to the authors). One of the strengths of open-access articles is also to promote that accessibility of data and code so that work can be re-produced, and data shared easily.

Response: Thank you for this comment. We now published the following data on Zenodo (doi: 10.5281/zenodo.3895297): - Thies disdrometer measurement outputs: daily .csv files. - OTT pluviometer measurement outputs: daily .csv files. - 2DVD measurement outputs: daily .sno files (containing the information of successfully matched hydrometeors) provided in ASCII format. - Metadata, i.e. user manuals and specifications for these 3 measurement instruments.

We have only used standard libraries (in the R software environment) for the processing of the data. Regarding the classification algorithm applied to 2DVD measurements, we were in close exchange with Joanneum Research and partly used empirical relationships derived by them, which we mention in the methodology section (2.2) as well as in the acknowledgements.

Changes in the manuscript (data availability section): The data used in this study, i.e. measurement outputs of the Thies disdrometer, the OTT pluviometer as well as the two-dimensional video disdrometer (2017-07-01–2019-06-30), can be found in Fehlmann et al. (2020).

---

## Author Comment (AC4) · 22 Jul 2020

Please find below a revised answer to comment 1.2:

Comment 1.2 Section 2: Did the Authors applied any filtering method to eliminate the so called "spurious drops" due to win, splashing, or mismatch? Several studies that used disdrometer measured DSD applied a filter criterion based on fall velocity such as the one adopted in Tokay et al. 2001 and valid only for rain.

Response: Thank you very much for this comment. Indeed, this effect exists and several studies apply filter algorithms to remove spurious measurements of mostly larger particles from 2DVD or other video disdrometer measurements by applying a filter based on the combined velocity-diameter information (e.g., von Lerber et al., 2017;

[Figure]

Raupach and Berne, 2015). However, as shown by Friedrich et al. (2013) for the Parsivel disdrometer such effects mostly occur at high wind speeds (exceeding 20 m/s). Although the effect of splashing might differ for different disdrometer types because of the different shape (in particular the two arms of the Parsivel vs. the two rails of the Thies disdrometer), Friedrich et al. (2013) is also cited in the context of Thies disdrometer analyses (e.g., Chen et al., 2016). As our study is extremely wind sheltered we did not see the need of applying such a filter in this study.

Furthermore, to investigate the potential effect of applying a similar filter than Chen et al. (2016) at our study site, we did the following exploratory analysis: We filtered out from the binned V-D data all particles corresponding to bins which have a centroid outside +-60% of the theoretically expected velocity using the empirical V-D relationship for all precipitation types (see Fig. 3, red bins are excluded). We then calculated total volume for all particles during the 2 years of measurement, assuming a spherical shape for simplicity. The percentage of the invalid particles volume to total volume is 2.3% for the whole time period and a little bit higher in summer (months Apr-Sept: 4.7%) than in winter (Oct-Mar: 1.8%). Although the application of such a filter might have an effect on PSD correction factors for small volumes, the effect of on precipitation intensity, which is the focus of the study, will be rather small. Also, we might filter out too many particles as the disagreement of the Thies disdrometer with respect to the OTT pluviometer for liquid precipitation will even increase.

Chen, B., Wang, J., & Gong, D. (2016). Raindrop size distribution in a midlatitude continental squall line measured by Thies optical disdrometers over East China. Journal of Applied Meteorology and Climatology, 55(3), 621-634.

Changes in the manuscript (section 2.1): Note that in some studies using optical disdrometer measurements, additional filters are applied to remove spurious measurements due to splashing or margin faller effects (e.g., Chen et al., 2016; Friedrich et al., 2013; Raupach and Berne, 2015; von Lerber et al., 2017). Usually, such filters are based on a validity check of the combined diameter and fall velocity information,

e.g. excluding data that are more than 60% above or below the fall velocity-diameter relationship for rain (Jaffrain and Berne, 2011). However, as investigated in detail by Friedrich et al. (2013) for Parsivel disdrometers, such spurious measurements mostly occur at wind speeds exceeding 20 m/s. As our study is extremely wind sheltered, we thus did not see the need of applying such a filter in this study. This is further supported by an exploratory analysis of applying the filter proposed by Chen et al. (2016) to the Thies disdrometer measurements over the full time period, which revealed that the volume contribution of the filtered particles is only very small (in the order of 2 – 3 %) in our case.

[Figure]

V [m/s]

D [mm]

**Fig. 1.**

[Figure]

---

## Author Response (AR2)

Dear Alexis Berne

Thank you very much for your final, valid concern and the following clarification of our questions. Please find below the revised answer to comment 1.2 of reviewer #1, which I also posted in the discussion of the article, as well as a version of the manuscript (text only) with marked changes.

In addition to changes mentioned in the revised comment #1.2 below, I have changed slightly the data availability section, now also including the LUFFT weather sensor data in the cited Zenodo repository. Also, I am happy to express my thanks to you, the anonymous reviewers and Adrien Guyot for the constructive feedbacks during the revision of this article in the acknowledgments section of the revised manuscript.

Kind regards,
Michael Fehlmann

**Revised reply to comment 1.2 of reviewer #1:**

**Comment 1.2** Section 2: Did the Authors applied any filtering method to eliminate the so called "spurious drops" due to win, splashing, or mismatch? Several studies that used disdrometer measured DSD applied a filter criterion based on fall velocity such as the one adopted in Tokay et al. 2001 and valid only for rain.

**Response:** Thank you very much for this comment. Indeed, this effect exists and several studies apply filter algorithms to remove spurious measurements of mostly larger particles from 2DVD or other video disdrometer measurements by applying a filter based on the combined velocity-diameter information (e.g., von Lerber et al., 2017; Raupach and Berne, 2015). However, as shown by Friedrich et al. (2013) for the Parsivel disdrometer such effects mostly occur at high wind speeds (exceeding 20 m/s). Although the effect of splashing might differ for different disdrometer types because of the different shape (in particular the two arms of the Parsivel vs. the two rails of the Thies disdrometer), Friedrich et al. (2013) is also cited in the context of Thies disdrometer analyses (e.g., Chen et al., 2016). As our study is extremely wind sheltered we did not see the need of applying such a filter in this study.

Furthermore, to investigate the potential effect of applying a similar filter than Chen et al. (2016) at our study site, we did the following exploratory analysis: We filtered out from the binned V-D data all particles corresponding to bins which have a centroid outside +-60% of the theoretically expected velocity using the empirical V-D relationship for all precipitation types (see Fig. 3, red bins are excluded). We then calculated total volume for all particles during the 2 years of measurement, assuming a spherical shape for simplicity. The percentage of the invalid particles volume to total volume is 2.3% for the whole time period and a little bit higher in summer (months Apr-Sept: 4.7%) than in winter (Oct-Mar: 1.8%). Although the application of such a filter might have an effect on PSD correction factors for small volumes, the effect of on precipitation intensity, which is the focus of the study, will be rather small. Also, we might filter out too many particles as the disagreement 
[revised manuscript text omitted]